# Characterization of the Cynomolgus Macaque Model of Marburg Virus Disease and Assessment of Timing for Therapeutic Treatment Testing

**DOI:** 10.3390/v15122335

**Published:** 2023-11-28

**Authors:** Elizabeth E. Zumbrun, Carly B. Garvey, Jay B. Wells, Ginger C. Lynn, Sean Van Tongeren, Jesse T. Steffens, Kelly S. Wetzel, Laura M. Gomba, Kristan A. O’Brien, Franco D. Rossi, Xiankun Zeng, Eric D. Lee, Jo Lynne W. Raymond, Diana A. Hoffman, Alexandra N. Jay, Elizabeth S. Brown, Paul A. Kallgren, Sarah L. Norris, Jean Cantey-Kiser, Humza Kudiya, Chris Arthur, Christiana Blair, Darius Babusis, Victor C. Chu, Bali Singh, Roy Bannister, Danielle P. Porter, Tomas Cihlar, John M. Dye

**Affiliations:** 1United States Army Medical Research Institute of Infectious Diseases, Frederick, MD 21702, USA; carly.b.garvey.ctr@health.mil (C.B.G.); jay.b.wells.ctr@health.mil (J.B.W.); ginger.c.lynn.ctr@health.mil (G.C.L.); sean.a.vantongeren.ctr@health.mil (S.V.T.); jesse.t.steffens.ctr@health.mil (J.T.S.); kelly.s.wetzel2.ctr@health.mil (K.S.W.); laura.gomba@nih.gov (L.M.G.); kristan.a.obrien.ctr@health.mil (K.A.O.); franco.d.rossi.civ@health.mil (F.D.R.); xiankun.zeng.civ@health.mil (X.Z.); eric.d.lee2@gmail.com (E.D.L.); jolynne.w.raymond.civ@health.mil (J.L.W.R.); diana.a.hoffman.mil@health.mil (D.A.H.); alexandra.n.jay.civ@health.mil (A.N.J.); elizabeth.s.brown34.ctr@health.mil (E.S.B.); paul.a.kallgren.ctr@health.mil (P.A.K.); sarah.l.norris2.civ@health.mil (S.L.N.); john.m.dye1.civ@health.mil (J.M.D.); 2Geneva Foundation, Tacoma, WA 98402, USA; 3PharPoint, Wilmington, NC 28405, USA; 4Gilead Sciences, Foster City, CA 94404, USA; humza.kudiya@gilead.com (H.K.); chris.arthur@gilead.com (C.A.); chris.blair@gilead.com (C.B.); darius.babusis@gilead.com (D.B.); victor.chu2@gilead.com (V.C.C.); bali.singh@gilead.com (B.S.); roy.bannister@gilead.com (R.B.); danielle.porter@gilead.com (D.P.P.); tomas.cihlar@gilead.com (T.C.)

**Keywords:** Marburg virus, filovirus, nonhuman primate, cynomolgus macaque, intramuscular, animal model, animal rule, telemetry, pathogenesis

## Abstract

Marburg virus (MARV) causes severe disease and high mortality in humans. The objective of this study was to characterize disease manifestations and pathogenesis in cynomolgus macaques exposed to MARV. The results of this natural history study may be used to identify features of MARV disease useful in defining the ideal treatment initiation time for subsequent evaluations of investigational therapeutics using this model. Twelve cynomolgus macaques were exposed to a target dose of 1000 plaque-forming units MARV by the intramuscular route, and six control animals were mock-exposed. The primary endpoint of this study was survival to Day 28 post-inoculation (PI). Anesthesia events were minimized with the use of central venous catheters for periodic blood collection, and temperature and activity were continuously monitored by telemetry. All mock-exposed animals remained healthy for the duration of the study. All 12 MARV-exposed animals (100%) became infected, developed illness, and succumbed on Days 8–10 PI. On Day 4 PI, 11 of the 12 MARV-exposed animals had statistically significant temperature elevations over baseline. Clinically observable signs of MARV disease first appeared on Day 5 PI, when 6 of the 12 animals exhibited reduced responsiveness. Ultimately, systemic inflammation, coagulopathy, and direct cytopathic effects of MARV all contributed to multiorgan dysfunction, organ failure, and death or euthanasia of all MARV-exposed animals. Manifestations of MARV disease, including fever, systemic viremia, lymphocytolysis, coagulopathy, and hepatocellular damage, could be used as triggers for initiation of treatment in future therapeutic efficacy studies.

## 1. Introduction

Marburg virus (MARV) causes severe disease in humans, with mortality reaching 23–90% [1]. MARV outbreaks likely result from spillover events from natural host reservoir species, such as the Egyptian fruit bat (*Rousettus aegyptiacus*) [2,3,4]. The first known outbreak of MARV occurred in 1967 in Marburg, Germany, and outbreaks have occurred sporadically since that time throughout sub-Saharan Africa [5,6,7,8,9,10,11,12]. Currently, no licensed MARV vaccines or therapeutics are available, and due to the sporadic nature of the outbreaks, animal models that recapitulate the human disease are needed for the development of these countermeasures.

MARV disease (MVD) in humans typically has an incubation period of 4–10 days and a mean of 9 days (range 8–16 days) between the onset of symptoms and death. Clinical signs of MVD in humans include fever, chills, myalgias, headache, encephalitis, and hemorrhagic manifestations progressing to severe coagulopathy, multiorgan system dysfunction, and shock. Clinical pathology findings include bone marrow suppression, systemic inflammation, hepatocellular damage, renal dysfunction, consumptive coagulopathy (with disseminated intravascular coagulation), vascular endothelial dysfunction, and hypovolemic shock. Anatomic pathology findings include lymphoid depletion of the spleen and lymph nodes, and necrosis and hemorrhage in numerous organs. Currently, although several MARV vaccine candidates are in early Phase 1–2 clinical trials [13,14,15], no vaccines or therapeutics for MVD have been approved.

For in vivo laboratory studies with MARV, cynomolgus macaques have been used for both vaccine and therapeutic studies [16,17,18,19,20,21,22] Historically, the rhesus macaque model has been used most frequently for testing potential filovirus antiviral therapeutics, whereas the cynomolgus macaque has been the predominant model for filovirus vaccine development studies, although these models may be used interchangeably in the future to provide a second model verification of vaccine or therapeutic efficacy under the Animal Rule. The disease manifestations caused by intramuscular (IM) inoculation with MARV in rhesus and cynomolgus macaques are historically very similar, with rhesus and cynomolgus macaques challenged by the IM route with 1000 plaque-forming units (pfu) MARV succumbing 7–8 days [23,24,25,26,27] or 6–10 days after inoculation, respectively [16,18,19,23,28,29,30,31,32,33]. Both the MARV rhesus and the MARV cynomolgus IM models are highly lethal, with manifestations including viremia, rash, coagulopathy, lymphopenia, elevated markers of inflammation, impaired liver and kidney function, and sustained fever. Both rhesus and cynomolgus models of IM MARV infection have been described in the literature as resulting in a disease course similar to that of humans; thus, either model is appropriate for Animal Rule studies [24,28,30,34,35].

In humans, MARV transmission may occur via contact (through broken skin or mucous membranes) with infectious blood, secretions, organs, or other body fluids of infected people, or through contact with materials contaminated with these fluids [34,35,36,37,38,39,40,41,42,43,44]. The majority of published data characterizing disease and mortality in cynomolgus macaques following exposure to MARV were obtained from studies involving IM injection [16,18,19,23,28,29,30,31,32,33]. The present study and two other recent MARV natural history studies [28,30] provide a wide breadth of historical data with which to characterize the performance of the IM cynomolgus macaque MVD model, providing evidence of consistent reproducibility in endpoints such as mortality over time from investigators at multiple institutions. The IM route of exposure is appropriate for Animal Rule studies involving the cynomolgus MARV model because it produces a consistent infection, key disease manifestations that closely resemble those occurring in fatal human cases, and uniform lethality. The high mortality in the cynomolgus macaque MVD model thus allows for the evaluation of MARV therapeutics that demonstrate a survival benefit, which is directly related to the desired clinical endpoint in humans.

Here we report a MARV natural history study, adherent to Good Laboratory Practices (GLP), undertaken to define the course of disease and provide a detailed assessment of the disease manifestations to enable use of the model in time-to-treat assessments of therapeutics under the Animal Rule. A total of 18 healthy, research-naïve cynomolgus macaques of Cambodian origin underwent surgical implantation of telemetry devices to continuously measure temperature and activity and central venous catheters (CVCs) to allow for blood collection in unanesthetized animals. A group of 12 animals (7 males, 5 females) was exposed by the IM route to a target dose of 1000 pfu MARV, and a group of 6 animals (4 males, 2 females) served as the mock-exposed control group. MARV-exposed and mock-exposed animals were housed in separate rooms within the same biosafety level 4 (BSL-4) suite but were otherwise subjected to the same experimental procedures.

In addition to morbidity and survival outcome, the primary endpoints determined through this study included the following assessments of MVD: clinical disease observations; detectable virus in plasma via reverse-transcription polymerase chain reaction (RT-PCR); detectable infectious viral load via plaque assay; alterations in serum chemistry, hematology, and coagulation parameters; and alterations in body temperature and activity by telemetry. Anatomic pathology analysis was also performed and gross pathology, histology, immunohistochemistry (IHC), and in situ hybridization (ISH) were reported. The results of these assessments enabled the determination of the time from MARV exposure to the onset, order of progression, frequency, and severity of disease manifestations.

## 2. Materials and Methods

### 2.1. Ethics Statement

Research was conducted under an Institutional Animal Care and Use Committee–approved protocol in compliance with the Animal Welfare Act, Public Health Service Policy, and other Federal statutes and regulations relating to animals and experiments involving animals. The facility where this research was conducted is accredited by AAALAC International and adheres to principles stated in the Guide for the Care and Use of Laboratory Animals, National Research Council 2011 [45].

### 2.2. Quality System

This study was conducted in accordance with GLP to align with regulatory guidance set forth by the United States Food and Drug Administration (FDA) per the Code of Federal Regulations (CFR) Title 21, Part 58. The study protocol was provided to the FDA for feedback prior to the initiation of the study, and the final report was provided to the FDA upon completion of the study.

### 2.3. Experimental Design

Experimentally naïve male and female cynomolgus macaques (*Macaca fascicularis*) of Cambodian origin from Worldwide Primates were used in this study. Animals were maintained at the United States Army Medical Research Institute of Infectious Diseases (USAMRIID) animal housing colony prior to assignment to the study and were transferred for acclimation to BSL-4 laboratory conditions 9 days prior to challenge. A total of 18 cynomolgus macaques were randomly assigned to either a mock-exposed group (n = 6 [2 females, 4 males]) or a MARV-exposed group (n = 12 [5 females, 7 males]), stratified by sex and balanced by weight, using a randomization plan created in SAS version 9.4. The study was not blinded. On Day 0, animals were 3.5–5.1 years of age and weighed 3.4–4.5 kg. Animals in the mock-exposed group were administered diluent (minimum essential medium with 2% heat-inactivated fetal bovine serum). Animals in the MARV-exposed group were exposed to a target dose of 1000 pfu MARV in diluent. Injections were administered on Day 0 (Table 1) via the IM route in the right quadriceps in a volume of 0.5 mL. The challenge dose of 1000 pfu of MARV is the dose most frequently used in published reports of the cynomolgus IM/MARV model [16,18,23,24,28,29,30,32,33,46].

### 2.4. Challenge Agent Source and Propagation History

MARV was isolated from an 8-month-old girl during an outbreak occurring in 2005 in Angola. The patient exhibited disease, was hospitalized, and died. The first passage of virus (designated virus seed pool 810820) was conducted at the Centers for Disease Control and Prevention using Vero E6 cells. A second passage of virus (designated WRC000041) was conducted at the University of Texas Medical Branch. WRC000041 was transferred to USAMRIID and propagated on BEI Resources Vero E6 cells to produce a passage 3 virus stock, the USAMRIID master seed stock, Lot R4410. The identity of this stock has been confirmed by agent-specific RT-PCR assay, as well as by sequencing on the Illumina MiSeq System (San Diego, CA, USA) (150 bp paired-end format). The full-length sequence of R4410 had 100% coverage and was found to be identical to the reference sequence (GenBank reference DQ447655.1) except for three bases in noncoding regions (DQ447655.1:g.19105A>T, DQ447655.1:g.19114A>C and DQ447655.1:g.28C>T). No bacterial or fungal contaminants were noted after incubation on chocolate agar or trypticase soy broth. Exclusionary RT-PCR data suggests that the stock is free from contamination by all viral agents tested. Mycoplasma levels were below the limit of detection (LOD) using the MycoAlert mycoplasma detection kit (Lonza; Bend, OR, USA). Endotoxin level, measured via limulus amebocyte lysate assay (QCL-1000 kit; Lonza), was below the LOD of the assay.

### 2.5. Clinical Observations

Animals were evaluated daily by study personnel according to the study schedule (Table 1), beginning 8 days prior to exposure and continuing through the end of the in-life phase. For observations and sample collections, a day began at 0000 and ended at 2359. At each awake observation, animals were assigned a responsiveness score using a 5-point scale as follows: 0 (alert, responsive, normal species-specific behavior), 1 (slightly diminished general activity, subdued, but responds normally to external stimuli), 2 (withdrawn, may have head down, upright fetal posture, hunched, reduced response to external stimuli), 3 (prostrate but able to rise if stimulated, or dramatically reduced response to external stimuli), and 4 (persistently prostrate, severely or completely unresponsive). The frequency of observations was adjusted based on responsiveness scores: when all animals were assigned a score of 0, observations were conducted twice daily; when any animal was assigned a score ≥ 1, observations of all animals were conducted five times per day, every 3–7 h. Observations were conducted prior to any blood collection events.

Animals were also evaluated for other signs of illness, which did not factor into euthanasia decisions, including but not limited to: swelling, rash, bleeding, and motor function. Other observations, such as biscuit/fruit consumption, condition of stool, and urine output, were also documented when possible.

Physical examination required animals to be anesthetized. Physical examination of animals under anesthesia occurred after awake observations on Day 0 (prior to exposure) and when an animal succumbed or was euthanized. Animals were anesthetized using 10 mg/kg ketamine IM. Anesthetized animals were examined for signs of illness, including rash, bleeding, discharge, swelling, lymphadenopathy, and inoculation site changes. Body weight measurements, collected without jackets, were recorded during anesthetized observations and at the time of euthanasia.

Other than scheduled anesthetized physical exams (on Day 0 or prior to euthanasia), all attempts were made to minimize instances in which animals were anesthetized. Animals were anesthetized for (a) blood collection when catheters were nonpatent (6 animals on one or more occasions each) or (b) replacement of damaged jackets (2 animals on one occasion each). Animals were not anesthetized on more than 3 consecutive days or when assigned a responsiveness score ≥ 2.

### 2.6. Body Temperature and Activity Monitoring by Telemetry

Body temperature and activity were monitored by telemetry using the Notocord-hem Evolution software platform (Version 4.3.0.77, Notocord Inc.; Le Pecq, France).

Temperature and activity signals were collected and analyzed as described previously using a sampling rate of 1 sample per second [47]. Baseline data were collected for 5 days prior to challenge (Days −6 through −2).

Sustained (>2 h) body temperature changes from baseline values, were identified according to the following definitions: significant increase or decrease in body temperature (a temperature value > 3 standard deviations above or below baseline); fever (a temperature value > 1.5 °C above baseline); hyperpyrexia (a temperature value > 3.0 °C above baseline); and hypothermia (a temperature value > 2.0 °C below baseline). In addition, the following body temperature parameters were assessed: maximum daily temperature elevation value (∆T_Max_, the largest ∆T value for the 24 h daily time period), daily percentage of significant temperature elevation values (TE_Sig_, the percentage of the 24-h daily time period during which body temperatures were significantly increased), and daily fever-hours (fever-h, the sum of the significant temperature increases, a representation of the intensity of fever).

When animals were found deceased, the time of death was estimated from the time at which the activity level dropped to an absolute minimum value, indicating an absence of any movement by the animal, coinciding with a time period showing a continuous and steady drop in body temperature, suggesting that the animal’s temperature control mechanism had ceased to function.

### 2.7. Clinical Pathology

Clinical pathology assessments were conducted on all animals to monitor progression of pathophysiological endpoints at multiple time points and at the time of euthanasia. Blood was collected from nonfasted animals via CVC (or venipuncture, when necessary, under anesthesia as described above) for hematology, coagulation, and clinical chemistry analyses according to the study schedule (Table 1). On scheduled blood collection days, blood collection occurred following the morning awake observation. For terminal blood collections, blood was collected by venipuncture or CVC under deep anesthesia (ketamine–acepromazine, ≥0.2 mL/kg IM) immediately prior to euthanasia.

#### 2.7.1. Hematology Analysis

Blood samples were collected in ethylenediamine tetraacetic acid (EDTA) tubes. Hematology analysis was conducted, within 4 h of sample collection, using an Abaxis Vetscan HM5 Hematology Analyzer (Abaxis; Union City, CA, USA). The following hematologic parameters were analyzed: neutrophils, lymphocytes, monocytes, white blood cell count, red blood cell count (RBC), hemoglobin, hematocrit, mean corpuscular volume, mean corpuscular hemoglobin, mean corpuscular hemoglobin concentration, RBC distribution width—standard deviation, RBC distribution width—coefficient of variation, platelet count, and mean platelet volume.

#### 2.7.2. Coagulation Analysis

Blood samples were collected in 3.2% sodium citrate tubes to obtain plasma within 4 h for prothrombin time (PT), activated partial thromboplastin time (APTT), fibrinogen, and D-dimer analyses using a Sysmex CA-1500 (Siemens; Washington, DC, USA).

#### 2.7.3. Blood Chemistry Analysis

Blood samples were transferred to serum tubes for alanine aminotransferase (ALT), aspartate aminotransferase (AST), alkaline phosphatase (ALP), total bilirubin, calcium, blood urea nitrogen (BUN), creatinine, total protein, albumin, glucose, uric acid, gamma glutamyl transferase (GGT), and amylase analyses within 5 h of sample collection, using a Piccolo Xpress (Abaxis), with the General Chemistry 13 reagent disk.

### 2.8. Plasma Viral RNA Assessment

Validated quantitative RT-PCR methods were used for the quantification of the systemic (plasma) concentration of viral RNA in samples collected at the indicated times (Table 1). Briefly, TRIzol LS-inactivated samples were extracted and eluted with AVE buffer using a QIAamp Viral RNA Mini Kit. The quantitative RT-PCR reaction used a master mix supplied by the Defense Biological Product Assurance Office (DBPAO). Samples were run on an ABI 7500 Fast Dx (Thermo Fisher; Waltham, MA USA), and data were acquired and analyzed using an ABI 7500 FAST SDS v1.4.

Individual PCR reactions were processed in a total volume of 20 μL containing the following assay parameters: 14.6 μL of master mix stock (U.S. Department of Defense, Joint Program Executive Office [Chemical, Biological, Radiological and Nuclear Defense], Joint Project Lead Enabling Biotechnologies, DBPAO), 0.4 μL SS II RT/Platinum Taq (Thermo Fisher), and 5 μL of extracted sample. Thermocycler conditions for the MARV DBPAO assay consist of the following cycles: Reverse transcription at 50 °C for 15 min, Taq activation at 95 °C for 5 min, 45 cycles of denaturation at 95 °C for 1 s, and annealing/extending at 60 °C for 26 s.

The endpoint of the assay was reported as genomic equivalents per PCR reaction (ge/rxn) of MARV Angola virus in test samples. The concentration of unknown samples was determined based on the standard curve run on the same plate. The standard curve fits a linear regression, y = mx + b, where y is the instrument response of each standard (Ct), x is the log of known concentrations of the standards (ge/rxn), m is the slope of the standard curve, and b is the y-intercept. The ABI software reports cycle threshold (Ct) values and calculates slope, R^2^ values, y-intercept, and test sample concentrations (ge/rxn) automatically.

For MARV RNA, samples with ≥2 of 3 replicates below the LOD (reported as “<LOD”) were imputed as 1.60 × 10^4^ ge/mL or 4.20 log_10_ ge/mL. The lower limit of quantitation (LLOQ) is 8.00 × 10^5^ ge/mL or 5.90 log_10_ ge/mL, and samples > LOD but < LLOQ were imputed as 8.00 × 10^5^ ge/mL or 5.90 log_10_ ge/mL. The upper limit of quantitation (ULOQ) is 8.00 × 10^11^ ge/mL or 11.90 log_10_ ge/mL, and samples greater than the ULOQ were imputed as 8.00 × 10^11^ ge/mL or 11.90 log_10_ ge/mL.

### 2.9. Plaque Assay for Challenge Backtiter and Viremia Assessment

Plaque assay was conducted as previously described to assess the challenge material and the burden of infectious virus in serum collected at multiple time points (Table 1) [48]. The LLOQ for the plaque assay is 3.00 log_10_ pfu/mL (1000 pfu/mL), equivalent to 10 pfu detected in two of two wells at the least dilute serum sample. The LOD for this plaque assay, starting with a 1:10 dilution of the serum and plating 100 µL per well in two wells (six-well dish format), is 1.70 log_10_ pfu/mL (50 pfu/mL), equivalent to 1 pfu detected in one of two wells. The ULOQ for the assay is 9.18 log_10_ pfu/mL. For infectious virus plaque serum viremia, values “>LOD, <LLOQ” were imputed as 1000 pfu/mL or 3.00 log_10_ pfu/mL, values “<LOD” were imputed as 50 pfu/mL or 1.70 log_10_ pfu/mL, and values > ULOQ were imputed as 9.18 log_10_ pfu/mL.

The titer of the diluted MARV material used to load the syringes for the MARV-exposed group was 2.25 × 10^3^ pfu/mL or 1125 pfu in 0.5 mL.

### 2.10. Euthanasia

To be declared moribund, an animal had to be assigned a responsiveness score of 4 (persistently prostrate, severely, or completely unresponsive). Animals assigned a responsiveness score of 4 were euthanized. The decision to euthanize an animal was made by trained and qualified personnel who had completed MARV-infected animal euthanasia training. Blood samples were collected as part of the terminal procedures for all animals. Unscheduled necropsies (on animals that were moribund or found deceased) were performed as soon as feasible following euthanasia or after they were found deceased, generally within 12 h of death. If the necropsy could not be performed immediately, the animal carcass was refrigerated to minimize tissue autolysis.

Mock-exposed animals were euthanized after anesthesia on Days 12 and 13 PI. No MARV-exposed animals survived to the end of the in-life phase. Scheduled necropsies were performed within 12 h of euthanasia. Terminal blood collection was taken by CVC or venipuncture. The final physical exam occurred prior to administration of the euthanasia solution.

CVC administration of a pentobarbital-based euthanasia solution (0.3–0.4 mL/kg) was performed under deep anesthesia with ketamine–acepromazine at a dosage of ≥0.2 mL/kg IM. Death was confirmed at ≥10 min after administration of barbiturate.

## 3. Results

### 3.1. Survival

All (n = 6) mock-exposed animals survived to the end of the in-life phase of the study (Figure 1). All (n = 12) MARV-exposed animals succumbed on Days 8–10 PI (Figure 1). Nine of these animals were euthanized when they were deemed moribund on Day 8 PI (n = 5, animals 9, 12, 13, 14, and 17), Day 9 PI (n = 3, animals 8, 15, and 18), and Day 10 PI (n = 1, animal 10). Three MARV-exposed animals were found deceased: animal 16 on Day 8 PI and animals 7 and 11 on Day 9 PI. Animal 18 was euthanized just before midnight (2352 h) on Day 8 PI, but death was confirmed just after midnight (0004 h) on Day 9 PI.

The average survival time (time from virus exposure to the time deceased) for MARV-exposed animals was 203.1 h (8.46 days) with a range of 180.1–234.9 h (7.51–9.79 days; Table 2). All mortalities are attributed to MARV exposure, and the difference in survival between the mock-exposed and MARV-exposed group is statistically significant (*p* < 0.001) by Fisher’s exact test.

### 3.2. Responsiveness

Daily maximum responsiveness scores are summarized in Figure 2; the time from virus exposure to a responsiveness score of four in MARV-exposed animals is shown in Table 3. All mock-exposed NHPs appeared healthy and normal throughout the study and received responsiveness scores of zero at all timepoints (Figure 2).

In MARV-exposed NHPs, the onset of a clinically observed reduction in behavioral activity (i.e., assignment of a responsiveness score of at least one) first occurred on Day 5 PI with 6 of 12 animals receiving responsiveness scores ≥ one during at least one observation event (Figure 2). By Day 6 PI, 9 of 12 animals had responsiveness scores ≥ one. All animals had responsiveness scores ≥ one on Day 7 PI. The average time from exposure to a score of one for MARV-exposed animals was 135.1 h (range 113.1–162.2 h). The average time that elapsed between the assignment of a responsiveness score of one and time deceased was 68.0 h (range 47.9–120.9 h). The time deceased is the time of confirmed death after euthanasia for animals receiving a responsiveness score of four, or the time of death as calculated by telemetry for animals found deceased. Though other cage-side observations were made, as described below, the only criterion for euthanasia was a responsiveness score of four.

### 3.3. Plasma Viral RNA

RT-PCR for MARV RNA was performed from the day of inoculation to Day 12 PI on plasma samples collected from each animal at intervals (Appendix A). Viral RNA was not detected at any time for any of the samples from the mock-exposed group (n = 6).

For greater granularity regarding the onset of the presence of plasma MARV RNA, both morning and evening samples were collected on Days 3 and 4 PI (except in mock-exposed animals 1 and 2 and MARV-exposed animals 17 and 18, due to nonpatent catheters). Viral RNA was first detected in plasma from MARV-exposed animals on Day 3 PI in three of the 12 animals (25%). While these plasma samples contained MARV RNA that was above the LOD, the amount detected was below the LLOQ. Of these 3 MARV-exposed NHPs with detectable virus in the plasma on Day 3 PI, 2 (animals 13 and 17) had detectable viral RNA in the morning sample and an additional animal (7) had detectable viral RNA in the evening sample.

By the morning of Day 4 PI, 11 of the 12 MARV-exposed NHPs had detectable viral RNA in plasma samples, with eight of these below the LLOQ and three in the quantifiable range (> LLOQ). The remaining MARV-exposed animal (10) that did not have RNA levels above the LOD on the morning of Day 4 PI had detectable levels of RNA in the evening. All MARV-exposed plasma samples collected for the remainder of the in-life period had viral RNA levels in the quantifiable range (Figure 3).

On Day 5 PI, 12 of the 12 MARV-exposed animals had quantifiable levels of plasma viral RNA, with an average of 7.68 log_10_ ge/mL and ranging from 6.15 to 9.57 log_10_ ge/mL (Appendix A). The levels of plasma viral RNA in the MARV-exposed animals continued to increase through Day 8 PI with an average of 10.78 log_10_ ge/mL and a range of 10.23–11.13 log_10_ ge/mL. These high levels of plasma viral RNA were sustained through Days 9 and 10 PI in the remaining animals, with averages of 10.61 and 10.49 log_10_ ge/mL, respectively. No samples measured were above the ULOQ.

Overall, peak RNA values in the MARV-exposed animals ranged from 10.23 to 11.31 log_10_ ge/mL and occurred on Day 6 (n = 3), Day 7 (n = 3), Day 8 (n = 3), Day 9 (n = 2), and Day 10 (n = 1) PI. The highest values of MARV RNA for 7 of the 12 MARV-exposed NHPs occurred at the final sample collected before the animals succumbed to the disease and ranged from 10.23 to 11.13 log_10_ ge/mL. For five animals, the highest values of MARV RNA occurred one day (4 of 12 animals) or two days (1 of 12) prior to the last sample collected before the animals succumbed and ranged from 10.84 to 11.31 log_10_ ge/mL. In these five NHPs, subsequent samples, after the peak, also contained similarly high levels of MARV RNA (ranging from 10.52 to 11.17 log_10_ ge/mL).

### 3.4. Serum Infectious Virus Load (Serum Plaque Assay)

An assessment of serum infectious virus was performed by plaque assay on samples obtained on Day 0; Days 3, 5, 7, 9, and 12 PI; and from samples obtained at the time of euthanasia. The presence of infectious virus was not observed for any serum samples collected from mock-exposed animals at any time point (Appendix A and Figure 4). No infectious virus was detected in MARV-exposed serum samples obtained on Day 3 PI. In serum samples collected from the MARV-exposed animals, infectious virus was detected on Day 5 PI in 12 of the 12 animals, with values ranging from 3.83 to 9.18 log_10_ pfu/mL. On Day 5 PI, all NHPs had serum infectious virus above the LLOQ, and one of the 12 NHPs (animal 17) had serum infectious virus that was above the ULOQ. By Day 7 PI, serum infectious virus had reached high levels (8.79–9.18 log_10_ pfu/mL) in 11 of 11 animals, with seven of 11 animals above the ULOQ. Serum was not collected from animal 17 on Day 7 PI because this animal had a nonpatent catheter and a responsiveness score of two and therefore could not be anesthetized for blood collection. Of note however, animal 17 had infectious virus in serum above the ULOQ on both Day 5 and Day 8 PI. On Day 8 PI, high values for serum infectious virus were sustained in six of six MARV-exposed animals (8.93–9.18 log_10_ pfu/mL), with four of six at levels above the ULOQ. Likewise, three of three MARV-exposed animals on Day 9 PI (9.03–9.18 log_10_ pfu/mL) and one of one animal on Day 10 PI (9.18 log_10_ pfu/mL) had sustained high levels of serum infectious virus. Peak serum infectious virus for MARV-exposed NHPs was reached on Day 5 PI (n = 1), Day 7 PI (n = 7), Day 8 PI (n = 1), Day 9 PI (n = 2), and Day 10 PI (n = 1).

### 3.5. Body Temperature and Activity by Telemetry

Body temperature and activity were assessed via telemetry. In these temporally fine-grained assessments, except where noted, “days” are defined as 24-h periods measured from the actual time of challenge rather than by calendar day. This is in contrast to all other evaluations in this report, in which “Day x PI” denotes calendar days.

#### 3.5.1. Body Temperature

During the 5-day baseline measurement period prior to virus exposure, all subjects in both groups displayed normal diurnal variations in body temperature, with increases during the light period and decreases during the dark period (Figure 5 and Appendix A). For all mock-exposed animals, these normal diurnal variations continued after exposure. No mock-exposed animal exhibited sustained fever, hyperpyrexia, or hypothermia during the study. Sustained, significant body temperature increases and decreases in the mock-exposed group were infrequent.

Among MARV-exposed animals, the normal fluctuations in body temperature observed during the baseline period continued for the first 3–4 days after challenge. All MARV-exposed animals exhibited sustained significant increases in body temperature (see Section 2.6) and sustained fever. The sustained, significant increases in body temperature started, on average, at 3.54 ± 0.87 days following challenge with a range from 3.08 to 4.52 days (range excludes one early possible outlier, animal 15, at 1.33 days). This statistically significant temperature increase lasted between 3.18 and 4.41 days (also excluding animal 15) before a period of temperature decrease. Of note, 7 of 11 animals (animal 15 excluded) had onset of significant temperature increase at 3 days after challenge, and 4 of 11 had onset at 4 days after challenge. Sustained fever started, on average, at 4.42 ± 0.91 days following challenge, with a range from 3.27 to 6.25 days. Notably, 9 of 12 animals had an onset of sustained fever before 5 full days following challenge had elapsed.

MARV-exposed animals also showed a disruption in the normal diurnal temperature pattern during the period of fever, as shown in the example in Figure 5 and for all animals in Appendix A. This appeared to occur between 3 and 5 days after challenge.

Sustained hyperpyrexia occurred in 9 of the 12 MARV-exposed animals with onset, on average, 5.79 ± 0.76 days following challenge (range 4.90–7.23 days). In the MARV-exposed group, the maximum temperature elevation occurred on Day 6 PI (Figure 6).

By the analysis of several parameters—∆T_Max_, TE_Sig_, and fever-h (see Section 2.6)—the onset of temperature increase in MARV-exposed animals began at Day 3 PI (calendar days) for at least two NHPs, which had values distinctly higher than those of the mock-exposed group (Figure 6). By Day 4 PI, ∆T_Max_, TE_Sig_, and fever-h values of 9 of the 12 MARV-exposed animals were elevated relative to those in the mock-exposed group (all MARV-exposed animals except 7, 8, and 10 for all three parameters). By Day 5 PI, all MARV-exposed animals had elevated ∆T_Max_, TE_sig_, and fever-h values relative to the mock-exposed animals.

All MARV-exposed animals exhibited a period of significant body temperature decrease in the terminal phase, starting, on average, at 8.02 ± 0.71 days after challenge (range 6.98–9.48 days). The time between the onset of this period of significant body temperature decrease and time deceased was an average of 10.8 h (range 1.7–26.6 h; n = 12; Figure 7). In nine of the 12 MARV-exposed animals, this body temperature decline reached a sustained period of hypothermia at 8.08 ± 0.72 days after challenge (range 7.35–9.63 days), on average, and continued through the time of euthanasia or the estimated time of death. The time between the onset of this period of hypothermia and time deceased was an average of 6.6 h (range of 3.7–11.5 h; n = 9; Figure 7).

#### 3.5.2. Activity

Prior to virus exposure, all subjects in both groups exhibited typical diurnal variations in activity, with increases during the light period and decreases during the dark period (Figure 8 and Appendix A). All mock-exposed animals continued to show these normal diurnal fluctuations in activity after exposure and throughout the study. No mock-exposed animal showed any significant decrease in activity during light or dark periods. Five of six mock-exposed animals exhibited transient increases in daytime and/or nighttime activity after exposure.

Five of the 12 MARV-exposed animals exhibited increases in activity during the light period up to Day 5 PI. Eleven of the 12 MARV-exposed animals displayed significant reductions in light period activity lasting for 1 day (4 of 11), 2 consecutive days (3 of 11), or 3 consecutive days (4 of 11) prior to euthanasia or the estimated time of death; animal 13 was the only MARV-exposed animal that did not exhibit a significant reduction in daytime activity. Activity changes during the dark period were less consistent across MARV-exposed animals.

### 3.6. Clinical Pathology

The clinicopathologic changes in the bloodwork for MARV-exposed animals are consistent with the effects of systemic inflammation, catecholamine-mediated changes, consumptive coagulopathy, and hypovolemic shock, all of which can be attributed to infection with MARV. Ultimately, the systemic inflammation, coagulopathy, and direct cytopathic effects of MARV all contributed to multiorgan dysfunction, organ failure, and death in all 12 of the MARV-exposed animals (Figure 9). Statistically significant changes in clinical pathology parameters described here are differences between mock- and MARV-exposed groups in change from baseline values (*p* ≤ 0.05).

Evidence for systemic inflammation in the MARV-exposed animals (Figure 10) includes elevated white blood cell numbers; a decrease in albumin, a negative acute phase protein; and an initial increase in fibrinogen, a positive acute phase protein. Specifically, neutrophils exhibited a moderate increase from baseline beginning on Day 4 PI, reaching a four-fold increase from baseline by Day 9 PI. After an initial mild decline in 10 of the 12 MARV-exposed animals, lymphocyte counts rebounded and doubled baseline by Day 8 PI. In the majority of the MARV-exposed animals, albumin decreased from baseline starting on Day 5 PI, with the decline reaching statistical significance on Days 6, 7, and 9 PI. Fibrinogen exhibited an upward trend on Days 4 and 5 PI prior to declining below baseline beginning on Day 7 PI. Gross and microscopic correlates of systemic inflammation included skin rash in 8 of 12 animals and elevated numbers of inflammatory cells (lymphocytes, macrophages, and neutrophils) present in the liver of all 12 animals.

Evidence for consumptive coagulopathy (Figure 10) includes markedly prolonged clotting times (PT and APTT) in addition to decreased fibrinogen, an increase in D-dimers, and an initial decrease in platelet counts. In particular, clotting times increased significantly above baseline in MARV-exposed animals beginning on Day 4 PI in 11 of 12 animals (APTT) and Day 6 PI in 10 of 10 animals (PT), increasing to double baseline for both tests by Day 7 PI. An upward trend in D-dimers began on Day 6 PI, with increases in 11 of 11 MARV-exposed animals by Day 7 PI, when the increase—at that point, four-fold above baseline—reached statistical significance. The increase in D-dimers continued to disposition. Platelet counts exhibited a mild downward trend (lacking statistical significance) in 10 of the 12 MARV-exposed animals from Days 4 through 5 PI before returning to normal reference intervals for cynomolgus macaques by disposition. This consumptive coagulopathy was associated with the hemorrhage identified in multiple tissues in MARV-exposed animals at disposition.

Increases in ALT, AST, and ALP, followed 1–2 days later by elevated total bilirubin and GGT (Figure 11), are all attributed to hepatocellular damage noted histologically. Specifically, a statistically significant increase in ALT concentration above baseline began on Day 6 PI, reaching up to 55-fold above baseline by disposition; this increase occurred in 10 of 10 MARV-exposed animals. AST concentration increases in 8 of the 12 MARV-exposed animals, which began on Day 5 PI and reached up to 44-fold above baseline by disposition, were statistically significant on Days 5, 6, and 7 PI. ALP concentrations began to rise on Day 4 PI in 9 of the 12 MARV-exposed animals, reaching statistical significance on Days 5, 6, 7, and 9 PI. GGT concentrations began to rise on Day 5 PI in several MARV-exposed animals and progressively increased until disposition; this increase was statistically significant on Days 6, 7, and 9 PI. Total bilirubin exhibited a mild increase on Day 6 PI in several MARV-exposed animals; this increase reached statistical significance on Day 7 PI when it was triple the baseline value. This pattern is consistent with initial hepatocellular damage, characterized by the release of hepatocellular enzymes, followed by downstream effects on the biliary system, evidenced by the subsequent increase in GGT and total bilirubin. The histopathologic correlate of hepatocellular damage and necrosis was present in all 12 MARV-exposed animals at disposition.

Moderate to marked elevations in BUN and creatinine (Figure 12), exhibited in 10 of the 12 MARV-exposed animals, were statistically significant on Days 7 and 9 PI. Both analytes increased to more than three-fold above baseline values by disposition. These changes are mainly attributed to pre-renal processes, including dehydration and hypotension, particularly given the lack of microscopic damage to the renal tissue itself.

### 3.7. Anatomic Pathology

Overall, gross, and microscopic findings suggest that MARV infection by IM inoculation caused lesions that included the following: hepatocellular degeneration and necrosis; lymphoid depletion and lymphocytolysis, especially in the spleen, gastrointestinal-associated lymphoid tissue, and lymph nodes; macular skin rash; hemorrhage in multiple organs, including the heart, urinary bladder, and testes; and degeneration, necrosis, and inflammation at the inoculation site. The hepatic degeneration and necrosis were likely a direct effect of the virus; this interpretation is supported by IHC- and ISH-positive signal in the hepatocytes. Changes in spleen, lymph nodes, and other lymphatic tissue can be attributed to MARV infection by their presence in all MARV-exposed animals and their absence in mock-exposed animals. The hemorrhage found in various tissues was likely caused by multifactorial and secondary downstream effects of the virus. Additional findings are summarized briefly below.

#### 3.7.1. Gross Findings

The disease-related gross findings exhibited most consistently by MARV-exposed animals included liver discoloration (pale, 12 of 12 animals), soft (friable) liver (11 of 12), liver enlargement (10 of 12), spleen enlargement (12 of 12), and a firm and/or pale spleen (11 of 12). Additionally, 8 of the 12 animals exhibited a skin rash, and 7 of 12 animals had red discoloration (suspected hemorrhage) in various organs. None of these findings was seen in the mock-exposed animals.

#### 3.7.2. IHC and ISH

All MARV-exposed animals had positive IHC and/or ISH results in the lung, heart, liver, spleen, kidney, inoculation site, adrenal gland, inguinal lymph node, small intestine, pancreas, brain, and eye (Figure 13 and Figure 14). All MARV-exposed females were IHC- and ISH-positive in the uterus and ovary. All MARV-exposed males were IHC- and ISH-positive in the testis and epididymis (Figure 14); the prostate was also IHC- and ISH-positive in the four males for which this tissue could be examined. Mock-exposed animals were negative for IHC and ISH across all tissues examined.

#### 3.7.3. Findings in Specific Tissues

This section provides a brief summary of changes in tissues that clearly differentiate between MARV- and mock-exposed animals.

##### Liver

All 12 MARV-exposed animals exhibited mild to marked hepatocyte degeneration and necrosis. Each MARV-exposed animal had minimal to moderate inflammation that expanded portal regions and/or occurred in areas of necrosis or surrounding larger blood vessels. Inflammatory leukocytes were composed of mononuclear cells with fewer neutrophils. No mock-exposed animal had hepatocyte degeneration or necrosis. Four of six mock-exposed animals had minimal hepatic infiltrates of histiocytes and lymphocytes.

##### Spleen

The spleen of each MARV-exposed animal was characterized by moderate to severe lymphoid depletion, mild to marked lymphocytolysis, minimal to moderate hemorrhage in the periarterial lymphoid sheaths, and mild to severe fibrin deposition in the red pulp. None of these changes was exhibited by mock-exposed animals.

##### Lymph Nodes

All 12 MARV-exposed animals demonstrated minimal to marked lymphoid depletion and lymphocytolysis in all lymph nodes examined. Some lymph nodes had increased levels of inflammatory infiltrates within the sinuses composed of mononuclear cells admixed in variable nodes and in variable animals with few neutrophils and eosinophils. No lymphoid depletion or lymphocytolysis was noted in any mock-exposed animal.

##### Gastrointestinal Tract

All 12 MARV-exposed animals exhibited minimal to moderate lymphoid depletion and lymphocytolysis in one or more areas of the gastrointestinal tract. Mock-exposed animals did not demonstrate lymphoid depletion or lymphocytolysis in the gastrointestinal tract.

##### Kidney

In the MARV-exposed group, 10 of 12 animals exhibited minimal to mild mineralization within renal tubules. In the mock-exposed group, five of six animals had minimal mononuclear interstitial infiltrates, and two animals had minimal tubular mineralization.

##### Inoculation Site

Widely varying microscopic changes were noted at the inoculation site in 6 of 12 MARV-exposed animals. These changes, which were generally minimal to mild, included degeneration and necrosis, inflammation, necrotic debris, edema, and myocyte regeneration. One mock-exposed animal exhibited mild skeletal muscle degeneration, minimal necrosis, and moderate mononuclear inflammation at the inoculation site.

##### Reproductive Organs

No significant findings—other than positive ISH and IHC signal, as described above—were observed in the ovary or uterus of any female MARV-exposed animal. In three of seven male MARV-exposed animals, the testes exhibited mild to marked hemorrhage. Mock-exposed animals had no notable findings in male or female reproductive organs.

**Figure 14 viruses-15-02335-f014:**
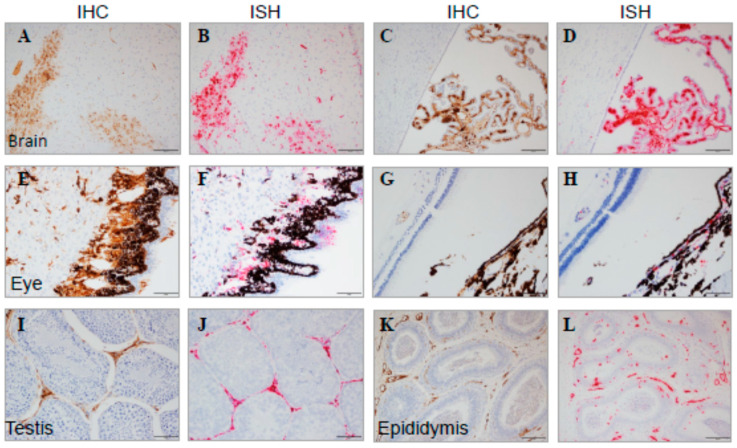
Immune-Privileged Tissues from MARV-Exposed Animals. Representative images of tissues from MARV-exposed animals are shown with IHC and ISH staining from adjacent sections. Brain (**A**): brain parenchymal IHC, (**B**): adjacent brain parenchymal ISH, (**C**): brain ventricles/choroid plexus IHC, (**D**): adjacent brain ventricles/choroid plexus ISH; eye (**E**): IHC at low magnification, (**F**): IHC at high magnification, (**G**): ISH at low magnification, (**H**): ISH at high magnification; testis (**I**): IHC, (**J**): ISH; and epididymis (**K**): IHC, (**L**): ISH. Positive staining from IHC tissue appears as a brown precipitate, and positive ISH staining is red. The scale bar is 50 μm.

## 4. Discussion

### 4.1. MARV Infection and Disease Kinetics

The primary objective of this study was to establish disease characteristics in cynomolgus macaques exposed to MARV via IM injection, including the time course and order of progression of the manifestations. After exposure to MARV, the disease progressed quickly (Figure 15) and all 12 animals (100%) succumbed within 7.51–9.79 days (n = 9 were euthanized; n = 3 were discovered deceased). Key disease manifestations included viremia, fever, macular skin rash, systemic inflammation, consumptive coagulopathy, hepatocellular degeneration and necrosis, lymphoid depletion and lymphocytolysis, and hemorrhage in multiple organs.

This section presents a day-by-day summary of the observed kinetics of infection and of these key disease manifestations in MARV-exposed animals from the time of virus exposure to the terminal phase of the disease. Additionally, key anatomic findings are summarized following the day-by-day summaries to corroborate, where possible, interpretations of clinical pathology alterations or to describe key disease findings identified through postmortem examinations.

#### 4.1.1. Day-by-Day Summary of Disease Manifestations

##### Day 0 through Day 2 PI

All MARV-exposed animals appeared healthy, and no abnormal clinical findings were noted during cage-side observations from Day 0 through Day 2 PI. Temperature remained normal in all MARV-exposed animals, except for one, which had an early significant temperature elevation on Day 1 PI. Diurnal temperature patterns remained normal for all animals during this period. There were no significant decreases in activity as measured by telemetry. No blood samples were collected on Days 1 or 2 PI and, although significant clinical changes were unlikely during this time, no conclusions can be made regarding possible clinical chemistry, hematology, or coagulation abnormalities or earlier onset of viremia that could have occurred during this time.

##### Day 3 PI

Although all MARV-exposed animals appeared clinically normal during cage-side observations on Day 3 PI, the onset of several disease manifestations was observed. At every observation, each animal was assigned a responsiveness score of zero (as defined in Section 2.5), and activity, as measured by telemetry, was not significantly decreased. However, six of 12 animals had statistically significant temperature increases, and onset of fever presented in two of these animals. Plasma viremia, assessed by RT-PCR, was detected (>LOD) in three of 12 animals, two with detectable levels in the a.m. samples and an additional animal with detectable viral RNA in the p.m. sample. While these RT-PCR values were above the LOD, they were below the LLOQ (i.e., not at quantifiable levels). One clinical pathology parameter was significantly altered in MARV-exposed animals on Day 3 PI: a statistically significant decrease from baseline in lymphocytes reflected the onset of a systemic inflammatory response.

##### Day 4 PI

Clinical signs of disease were not yet outwardly visible on Day 4 PI as all animals appeared clinically normal and received responsiveness scores of zero during all cage-side observations. Likewise, no statistically significant reductions in telemetry-based activity occurred. However, on Day 4 PI, onset of disease was detected in all MARV-exposed animals by several measures. In particular, statistically significant temperature increases (as defined in Section 2.6) were seen in 11 of 12 animals, and eight of 12 animals exhibited fever. At the morning assessment of viremia by plasma RT-PCR, 11 of 12 animals had detectable viral genomes. By the evening, 12 of 12 animals had detectable viral genomes within the plasma, and five animals had viral RNA values above the LLOQ. Lymphocyte counts remained depressed in 11 of 12 animals, indicating a continued systemic inflammatory response. The onset of coagulopathy occurred on Day 4 PI with a statistically significant elevation of APTT impacting 11 of 12 animals.

##### Day 5 PI

The onset of clinically observable manifestations of MARV disease first occurred on Day 5 PI in 6 of 12 animals, with maximum daily responsiveness scores (assigned during cage-side observations) of one (n = 4) or two (n = 2). While responsiveness scores indicated visibly reduced activity in 6 of 12 animals, only 1 of 12 animals had significantly reduced daytime activity as measured by telemetry. Other notable abnormalities recorded in cage-side observations included impaired motor function in 4 of 12 animals and no consumption of biscuits in 4 of 12 animals. Statistically significant temperature increases persisted in 12 of 12 animals, and 11 of 12 animals had fever. The onset of hyperpyrexia occurred in 4 of 12 animals, indicating an intensification of fever in these animals. Viremia, as detected by RT-PCR of plasma, increased, with quantifiable values detected in 12 of 12 animals at an average of 7.68 log_10_ ge/mL (range 6.15–9.57 log_10_ ge/mL). The onset of infectious virus in serum was detected by plaque assay in 12 of 12 animals with an average of 6.79 log_10_ pfu/mL (range 3.83–9.18 log_10_ pfu/mL). Clinical pathology parameters demonstrated the persistence and intensification of the inflammatory state with persistently decreased lymphocyte counts, onset of decreased albumin, and onset of increased fibrinogen. Coagulopathy persisted with increased APTT, and significantly elevated AST and ALP suggested the onset of hepatocellular damage.

##### Day 6 PI

MARV disease progressed on Day 6 PI, with 9 of 12 animals receiving maximum daily responsiveness scores of 1 (n = 5) or 2 (n = 4). Other abnormalities noted during cage-side observations included impaired motor function (5 of 12 animals), no consumption of biscuits (9 of 12), and rash (3 of 12). A significant reduction in daytime telemetry activity was observed in 4 of 12 animals. By Day 6 PI, all 12 animals had fever, and 7 of 12 animals exhibited hyperpyrexia. Plasma viremia, assessed by RT-PCR, showed a >2 log_10_ increase from the previous day with an average of 10.14 log_10_ ge/mL (range 8.93–11.31 log_10_ ge/mL). The inflammatory state persisted (as shown by decreased lymphocyte counts and decreased albumin), and indications of coagulopathy intensified with a significant rise in APTT, PT, and D-dimers over baseline. Hepatocellular damage became more overt with significant elevations in AST, ALT, ALP, and GGT.

##### Day 7 PI

Onset of responsiveness scores of three were first assigned to 2 of the 12 MARV-exposed animals on Day 7 PI. This was also the first day on which all animals (12 of 12) received a responsiveness score ≥ one. Notably, 8 of 12 animals had rash, 12 of 12 animals did not eat any biscuits, and 8 of 12 exhibited impaired motor function. Reduced daytime activity by telemetry was found in 9 of 12 animals. And though fever persisted in 11 of 12 animals, at some point during the course of the day, a significant temperature decrease occurred in 2 of 12 animals, and one of these animals became hypothermic. Plasma viremia was sustained at high levels with an average of 10.69 log_10_ ge/mL (range 10.00–11.17 log_10_ ge/mL) and serum live virus assays yielded 9.08 log_10_ pfu/mL (range 8.79–9.18 log_10_ pfu/mL). Aside from the continued inflammatory state (decreased lymphocyte counts and albumin), consumptive coagulopathy became progressively more severe with significantly decreased fibrinogen and further increases in APTT, PT, and D-dimers. Hepatocellular damage continued with significant elevations in ALT, AST, ALP, GGT, and bilirubin. BUN and creatinine also were significantly elevated over baseline values, reflecting the impairment of renal function.

##### Day 8 PI

Day 8 PI marked the first day on which MARV-exposed animals (6 of 12) succumbed to infection. Five animals met criteria for euthanasia (a responsiveness score of four), and one animal was found deceased with the time of death determined through telemetry. A seventh animal was assigned a score of four near the end of the day and was euthanized, but death was not confirmed until after midnight the following day. The five remaining animals had maximum responsiveness scores of two. Reduced daytime activity was observed in six of six animals on Day 8 PI. Fever continued until the onset of temperature decreases, which occurred in 9 of 12 animals on Day 8 PI with a total of seven animals developing hypothermia. Viremia remained at high levels with an average of 10.78 log_10_ ge/mL (range 10.23–11.13 log_10_ ge/mL) in plasma assessed by RT-PCR and 9.12 log_10_ pfu/mL (range 8.93–9.18 log_10_ pfu/mL) in serum assessed by plaque assay. Albumin continued to decrease on Day 8 PI, indicating acute inflammation, and lymphocytes became elevated, which could be due to inflammation or a catecholamine response. Coagulation parameters (APTT, PT, and D-dimers) were elevated even further on Day 8 PI, as were liver parameters (ALT, AST, ALP, GGT, and bilirubin) and kidney parameters (BUN and creatinine). These clinical pathology endpoints are compatible with consumptive coagulopathy, hepatocellular damage, and renal impairment.

##### Days 9–10 PI

Five of the six remaining MARV-exposed animals succumbed on Day 9 PI (including one for which euthanasia was administered the previous day but death confirmed very early on Day 9 PI). Of these animals, two were found deceased and three were euthanized after receiving a responsiveness score of four. The final surviving animal had a maximum responsiveness score of three and reached a score of four and was euthanized the following day (Day 10 PI). All remaining animals continued to have significantly reduced daytime activity by telemetry. By Day 9 PI, all remaining animals were in a state of significantly reduced body temperature and most (8 of 12) animals had reached a state of hypothermia in the final stage of disease. All animals displayed rash during the final stages of disease (observed cage-side and/or during the final anesthetized exam). Plasma RT-PCR values remained high in all animals on Days 9 and 10 PI (average of 10.61 and 10.49 log_10_ ge/mL, respectively). Live virus in serum remained similarly elevated, with an average of 9.09 log_10_ pfu/mL on Day 9 PI (9.03–9.18 log_10_ pfu/mL) and a value of 9.18 log_10_ pfu/mL on Day 10 PI. Clinical pathology parameters measured on Days 9 and 10 PI—indicating consumptive coagulopathy, hepatocellular damage, and renal impairment—remained altered (significantly on Day 9 PI), similar to Day 8 PI.

### 4.2. MVD Manifestations as Triggers for Treatment or for a Delayed Time-to-Treat Approach

Manifestations of MVD described here could be used as indicators for the initiation of treatment in future studies of investigational therapeutics using this disease model. The disease parameters identified herein could be utilized, either alone or in combination, as “real-time” triggers for the initiation of treatment in individual animals. Alternatively, a more feasible “delayed time-to-treat” approach could be applied to the treatment group as a whole based on significant disease manifestations identified in this report. In the latter case, treatment would be initiated on the same day for all animals in the treatment group, and the pertinent parameters would be retrospectively analyzed to verify whether the disease manifestation(s) occurred prior to treatment initiation.

Several factors guided the identification of disease manifestations that could serve as ideal treatment triggers. First, a potential trigger should be uniformly altered in all MARV-exposed animals. Furthermore, the potential trigger should exhibit changes that are statistically significant when compared with the MARV-exposed group baseline and with the mock-exposed group. Disease parameters that could serve as triggers should be objectively and quantitatively measurable. Importantly, disease manifestations that serve as treatment triggers must be detectable sufficiently early in the disease course, before the occurrence of potentially irreversible pathophysiological disease states.

On Day 3 PI, some animals began to show signs of disease, with onset of significant temperature increase in 6 of 12 animals, and detectable MARV RNA by RT-PCR in 3 of 12 animals. However, none of the disease parameters showed uniform changes across all animals on Day 3 PI. Therefore, a delayed treatment administered beginning on Day 3 PI would be prior to uniform onset of disease in this model. MARV-exposed animals succumbed starting on Day 8 PI, and animals displayed severe disease by numerous measures on Day 7 PI, with some animals reaching a responsiveness score of three and exhibiting more pronounced consumptive coagulopathy, hepatocellular damage, and renal dysfunction. As such, Day 7 PI is not a feasible treatment initiation day because the disease manifestations by this point are likely irreversible.

With Day 3 PI serving as an early boundary and Day 7 PI as a late boundary, objective parameters with onset of uniform, significant changes occurring between Days 4 and 6 PI were considered as possible “triggers to treat” or for “delayed time-to-treat” analysis. Among the parameters objectively assessed on Days 4, 5, and 6 PI, body temperature, MARV systemic viremia, lymphocyte count, APTT, AST, and ALP exhibited uniform (or nearly uniform in the case of lymphocyte count) changes in MARV-exposed animals (see Table 4 and Figure 16). Body temperature, lymphocyte count, and APTT exhibited statistically significant changes from baseline—and RT-PCR reached detectable, followed by quantifiable levels—on Days 4, 5, and 6 PI. Therefore, these parameters could be used as treatment triggers, or Days 4, 5, and 6 could be used as early, intermediate, or late treatment initiation times. AST and ALP showed significant changes on Days 5 and 6 PI; thus, these parameters could also serve as treatment triggers or could justify Day 5 PI (intermediate) or Day 6 PI (late) as predetermined treatment initiation times.

#### 4.2.1. Body Temperature

On Day 4 PI, 11 of the 12 MARV-exposed animals had statistically significant temperature elevations over baseline as measured by telemetry (Table 4 and Figure 16). For 8 of 12 animals, the temperature increase qualified as a “fever” (see Section 2.6). In contrast, no mock-exposed animals had any episodes of fever throughout the challenge period. Two mock-exposed animals did exhibit significant temperature elevations, but these were transient (less than 2 h in duration); thus, fever, as defined here, is a more specific and reliable indicator of disease in this model. The majority (9 of 12) of the MARV-exposed animals also had greater T_Max_, TE_Sig_ duration, and fever-h than animals in the mock-exposed group on Day 4 PI (Figure 6).

On Day 5 PI, all MARV-exposed animals (12 of 12) exhibited significantly increased body temperature, and 11 of 12 had fever. Hyperpyrexia was seen in 5 of 12 animals on Day 5 PI, but because hyperpyrexia was never seen in 3 of the 12 MARV-exposed animals throughout the entire study duration, this is not an ideal treatment trigger. All MARV-exposed animals (12 of 12) also had greater T_Max_, TE_Sig_ duration, and fever-h than animals in the mock-exposed group on Day 5 PI (Figure 6).

On Day 6 PI, all MARV-exposed animals (12 of 12) met the criteria for fever, and hyperpyrexia occurred in 7 of 12 animals. Additionally, more distinction was seen in T_Max_, TE_Sig_ duration, and fever-h in all MARV-exposed animals compared to mock-exposed animals (Figure 6).

Significant temperature elevation, present in 11 of 12 MARV-exposed animals on Day 4 PI, could be used as a treatment trigger, with the caveat that occasional temperature increases may not be specific to MARV infection. Of all the calculated measures of body temperature elevation determined through telemetry, fever is the most uniformly affected body temperature measurement that is also specific only to MARV-exposed animals. Thus, fever could serve as a reliable treatment trigger, or as a delayed time-to-treat disease manifestation, confirmed retrospectively. Day 5 PI is the earliest reasonable time at which fever could be used as a treatment trigger or delayed time-to-treat parameter because fever was present in all but one of the MARV-exposed animals on Day 5 PI, and all animals on Day 6 PI.

#### 4.2.2. Systemic Viremia

On Day 4 PI, MARV plasma viremia was assessed by RT-PCR at both a.m. and p.m. time points. At the morning time point, 11 of the 12 MARV-exposed animals had detectable viral genomes and, by the evening, the remaining MARV-exposed animal also had detectable plasma viral RNA (Table 4 and Figure 16). Only 5 of 12 MARV-exposed animals had viral genomes in the plasma that were in the quantifiable range. Nonetheless, since none of the mock-exposed animals had detectable virus in the plasma at any time, signals between the LOD and LLOQ could be used as a treatment trigger. As such, Day 4 PI is the earliest point at which this parameter could be used as a treatment trigger or delayed treatment approach.

By Day 5 PI, 12 of the 12 MARV-exposed animals had plasma viral RNA levels in the quantifiable range (Table 4) with an average of 7.68 log_10_ ge/mL (ranging from 6.15 to 9.57 log_10_ ge/mL; Appendix A and Figure 16). Because plasma viremia for all animals is quantifiable, and more severe, on Day 5 PI, this would be a more rigorous time to use RT-PCR values as a treatment indication and would be less likely to be impacted by any unforeseen assay variability.

The amount of viral genome detected in the plasma increased over 2 log_10_ by Day 6 PI, with an average of 10.14 log_10_ ge/mL (range 8.93–11.31 log_10_ ge/mL; Appendix A and Figure 16). Considering the magnitude of increase in viremia at Day 6 PI, approximately 2 days after onset of detectable viremia, and approximately 2 days between these high values and the time at which animals start to succumb from infection, Day 6 PI can be considered a late time for treatment.

#### 4.2.3. Lymphocytolysis

Lymphocytes were depressed in the MARV-exposed group relative to the mock-exposed group on Day 4 PI (*p* = 0.006 comparing MARV vs. mock change from baseline) with 11 of 12 animals having reduced lymphocyte counts relative to their baseline values (Table 4 and Figure 16). This trend, indicative of lymphocytolysis, started on Day 3 PI (*p* = 0.004) and continued through Day 5 PI (*p* = 0.003) and Day 6 PI (*p* = 0.013). Thus, lymphocyte counts are the earliest significant change that could be used, perhaps in combination with other clinical parameters, as a treatment trigger.

#### 4.2.4. Coagulopathy

Although alterations of several coagulation parameters were indicative of consumptive coagulopathy late in the MVD course, APTT was the earliest of these to demonstrate a significant change from baseline (Table 4 and Figure 16). On Day 4 PI, APTT was significantly elevated in MARV-exposed animals (*p* = 0.008 comparing MARV vs. mock change from baseline) with increases observed in 11 of 12 animals. By Day 5 PI, the increase in APTT was enhanced with 12 of 12 animals impacted (*p* = 0.002), and levels rose further on Day 6 PI (*p* = 0.016). Therefore, APTT change from baseline measurements on Days 4, 5, or 6 PI could be a useful added parameter for determining the treatment trigger, or retrospectively confirming the active MARV-associated disease state at the time of treatment.

#### 4.2.5. Hepatocellular Damage

Alterations in hepatic parameters were among the later changes to occur in this MARV model. However, two hepatic-related parameters, AST and ALP, demonstrated significant elevations by Day 5 PI (*p* = 0.002 and 0.008, respectively; Table 4 and Figure 16). By Day 6, PI, increases in AST and ALP were observed in all animals (10 of 10 animals tested) and were significantly elevated (*p* = 0.006 for both parameters). Increases in AST and ALP over baseline on Day 5 and Day 6 PI are thus potentially useful parameters either as triggers or in support of this timeframe as a treatment window. More definitive markers of hepatocellular damage were apparent by Day 6 PI with significant increases in ALT and GGT.

## 5. Conclusions

This study report details the natural history of IM MARV infection in cynomolgus macaques, conducted in accordance with the FDA CFR Title 21, Part 58 GLP. The objective of this study was to establish the characteristics of disease in MARV-exposed cynomolgus macaques relative to mock-exposed animals and identify key manifestations of disease for use as potential triggers for the initiation of treatment in subsequent evaluations of investigational therapeutics.

### Comparison to Two Other MARV Natural History Studies

Two other groups have also recently published well-characterized MARV natural history studies in cynomolgus macaques to satisfy the Animal Rule (Cromer et al. [30] and Alfson et al. [28]). Indeed, well-characterized animal models are needed at multiple institutions where vaccines and therapeutics will be tested to satisfy FDA’s Animal Rule approval pathway.

A comparison of the present study with the Comer et al. and Alfson et al. studies reveals numerous similarities among the studies as well as some key differences (Table 5) [28,30]. All three studies used a quality system such as GLP to investigate the disease course in male and female cynomolgus macaques following IM inoculation of a target dose of 1000 pfu of MARV Angola. All three studies definitively demonstrated clinical changes known to be associated with MARV infection of humans and NHPs, such as fever, systemic viremia, lymphocytolysis, coagulopathy, and hepatocellular damage [34,49,50,51]. MARV infection was fully lethal in all three studies and all mock-exposed animals remained healthy throughout the studies.

While the overall timing of clinical changes was similar in all three studies, serum samples were plaque assay-positive as early as Day 2 PI for Alfson et al. and Day 3 PI for Comer et al., and the plaque assay was more sensitive than the RT-PCR assay for measuring circulating virus in these studies [28,30]. In contrast, in the study reported herein, Day 3 serum samples were negative for virus in the plaque assay, but plasma samples yielded detectable MARV by RT-PCR starting between Days 3 and 4 PI.

One important difference between the study described here and those of Alfson et al. and Comer et al. [28,30] is the use of CVCs to collect blood in the present study; this minimized the need for anesthesia events during the in-life phase of the study. Furthermore, animals did not undergo any anesthesia events during the “critical phase”, when illness was progressing in this study. The inclusion of CVCs facilitated more frequent blood collection events without the need to repeatedly anesthetize critically ill animals. While Bennett et al. [52] showed no difference in outcome after several anesthesia events (on days 2, 4, and 6 PI) in EBOV-infected rhesus macaques, it is not known whether the anesthesia schedule used in the Alfson et al. and Comer et al. studies impacted the duration of the disease (i.e., time to death or euthanasia) for MARV-challenged cynomolgus macaques.

Second, the studies differed in the euthanasia criteria employed. The study reported herein euthanized animals based only on the responsiveness of the animal. The use of such stringent euthanasia criteria helps ensure that animals are not euthanized at a point when they could still recover; this is particularly important when a model is used for a therapeutic study with late treatment initiation, wherein animals may already be sick when treatment is initiated.

The median survival times in Alfson et al. (7.95 days PI) [28] and Comer et al. (7.3 days PI) [30] were shorter than in the study reported here (median = 8.33 days) (Table 5). Experimental design factors that could have contributed to the longer survival time in this study include the use of CVCs, eliminating the need for anesthesia events when animals were critically ill, or the use of more stringent euthanasia criteria in this study. Other factors that could have contributed to differences in survival time include differences in the origin of the cynomolgus macaques, the virus stocks used, or the animals’ age or weight (Table 5).

Finally, the study reported here included blood collection on three key days (Days 4, 5, and 6 PI) that would likely be used in this model for time-to-treat evaluation of therapeutic efficacy, whereas the other two groups measured clinical parameters on Days 3, 5, and 7 PI, but not on Days 4 and 6 PI. The inclusion of timepoints on all days when disease manifestations appear (Day 3 PI onward) allows for a thorough characterization of the treatment window. Here, we provide a thorough data set demonstrating the clinical changes occurring throughout the time period that could be used for decisions on the initiation of MARV therapeutic treatment in a future study under the Animal Rule.

## Figures and Tables

**Figure 1 viruses-15-02335-f001:**
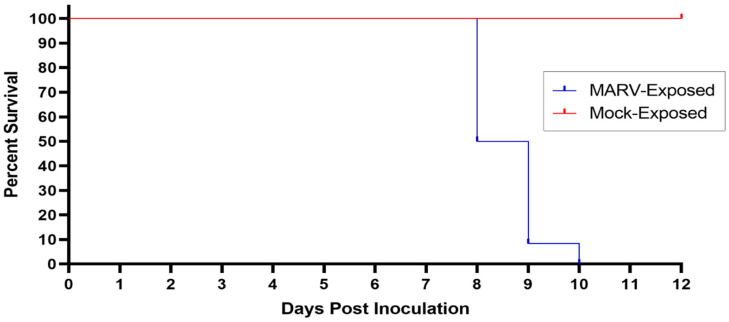
Kaplan–Meier Plot of Survival of MARV-Exposed and Mock-Exposed Cynomolgus Macaques. The survival of MARV-exposed (blue) and Mock-exposed (red) cynomolgus macaques is plotted on a Kaplan–Meier plot. All (n = 12) MARV-exposed animals succumbed between Days 8 and 10 PI. All (n = 6) mock-exposed animals survived until the end of the study.

**Figure 2 viruses-15-02335-f002:**
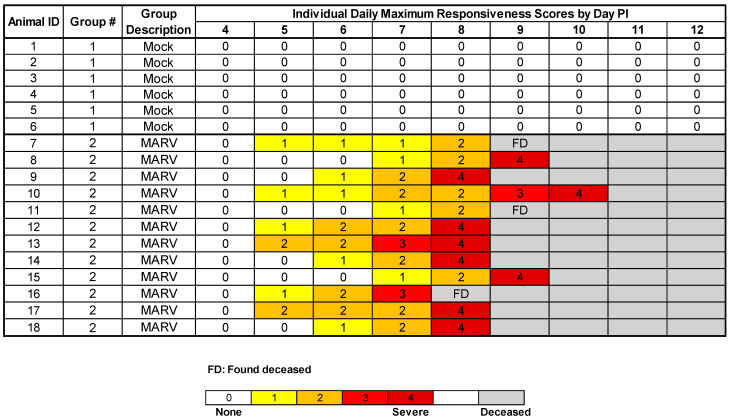
Daily Maximum Responsiveness Scores. The maximum responsiveness score for each animal on each day is displayed. At each awake observation, animals were assigned a responsiveness score using a five-point scale as described in the Methods. Animals were euthanized when they reached a responsiveness score of four. FD = Found Deceased.

**Figure 3 viruses-15-02335-f003:**
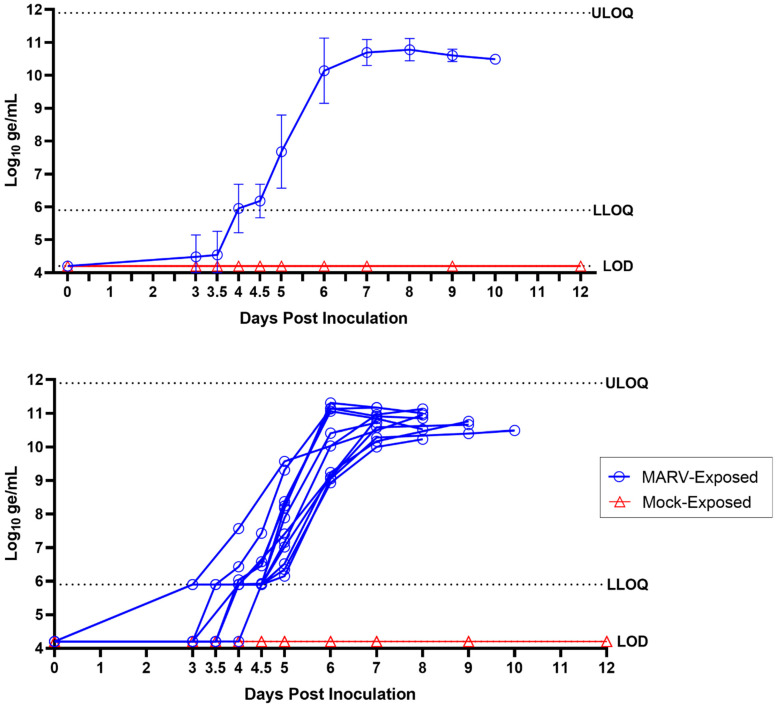
Plasma MARV RNA (RT-PCR). Top panel shows group means; error bars represent standard deviation. Bottom panel shows values for individual animals. LLOQ = 5.90 log_10_ ge/mL; LOD = 4.20 log_10_ ge/mL; ULOQ = 11.90 log_10_ ge/mL.

**Figure 4 viruses-15-02335-f004:**
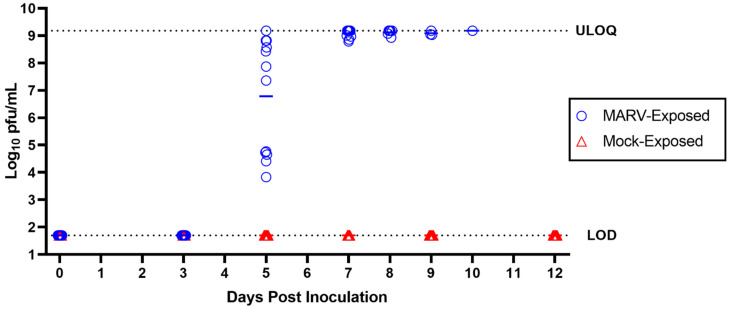
Serum Infectious Virus. MARV plaque assays were performed on serum samples collected from mock-exposed and MARV-exposed animals at intervals throughout the study. LOD = 1.70 log_10_ pfu/mL; ULOQ = 9.18 log_10_ pfu/mL. Horizontal bars represent group means.

**Figure 5 viruses-15-02335-f005:**
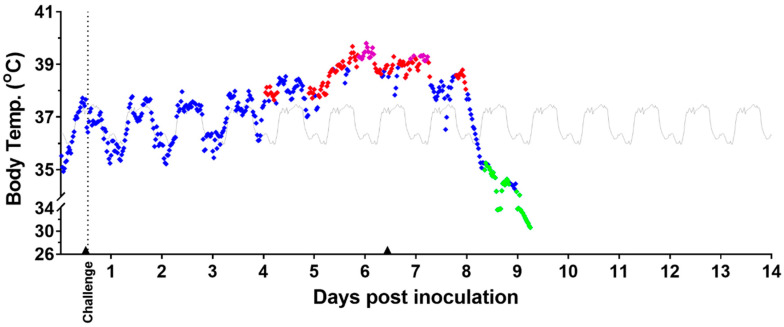
Body Temperature Measured by Telemetry—Representative Example. Temperature readings assessed by telemetry for a representative MARV-exposed animal are shown above. Analysis of body temperature (°C). Body temperature values displaying normal (♦), fever (♦), hyperpyrexia (♦), and hypothermia (♦); baseline average values are in gray (▬). Anesthesia use (▲). Days post inoculation = calendar days.

**Figure 6 viruses-15-02335-f006:**
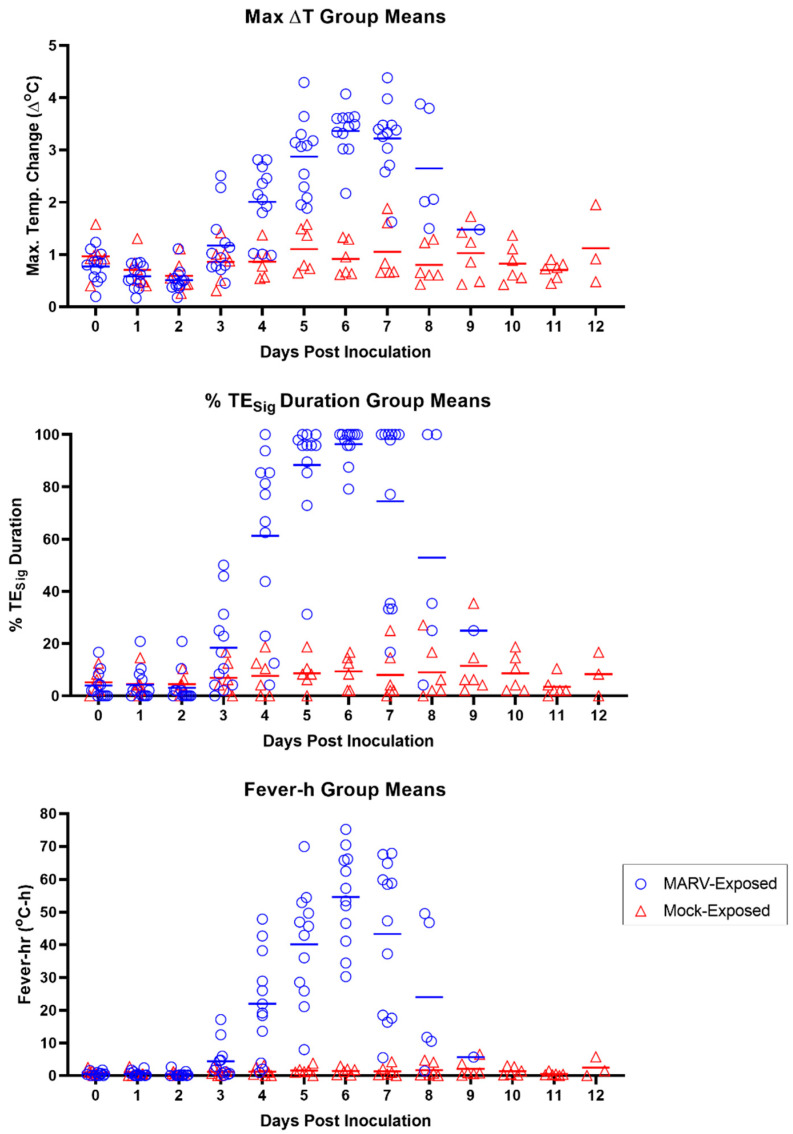
Body Temperature Measured by Telemetry. Results of the individual MARV-exposed (blue open circles) and mock-exposed (red open triangles) animals for the maximum daily temperature elevation value (∆TMax; top panel), daily percentage of significant temperature elevation values (TE_Sig_; middle panel), and daily fever-hours (fever-h; bottom panel). Days post-inoculation = calendar days. Horizontal bars in all panels represent group means.

**Figure 7 viruses-15-02335-f007:**
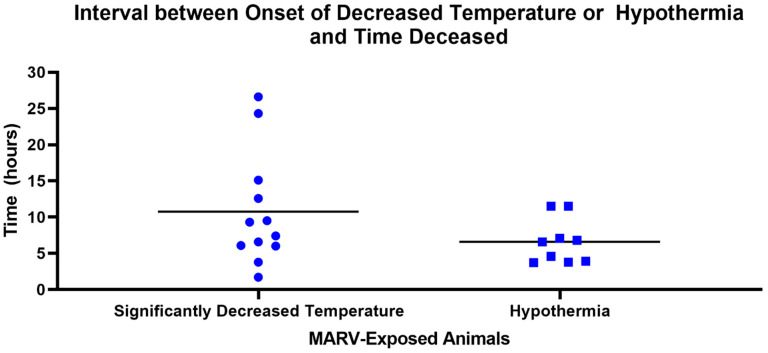
Interval between Onset of Decreased Temperature or Hypothermia and Time Deceased in MARV-Exposed Animals. Horizontal bars represent means. Significantly decreased temperature: temperature greater than 3 standard deviations below mean values calculated from the baseline period for longer than 2 h. Hypothermia: greater than 2.0 °C below baseline for longer than 2 h.

**Figure 8 viruses-15-02335-f008:**
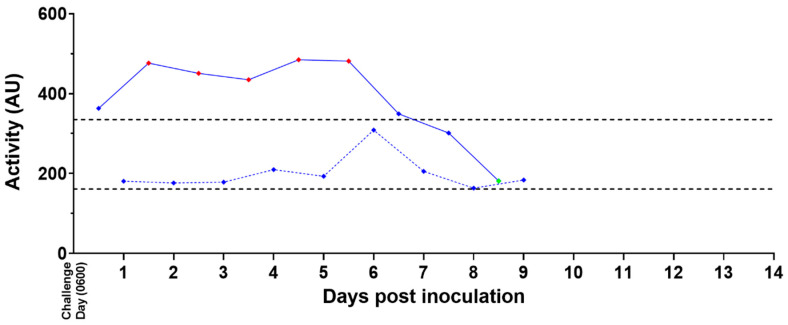
Activity Measured by Telemetry—Representative Example. Activity readings assessed by telemetry for a representative MARV-exposed animal are shown above. Values + 3 SD (♦) or −3 SD (♦) from baseline are statistically significant; values < 3 SD (♦) are not significant. Daytime (0600–1800 h) values are shown with solid blue lines (**──**♦**──**); nighttime (1800–0600 h) values are shown with dotted blue lines (**- - -**♦**- - -**). Baseline average values for daytime and nighttime are shown as black dotted lines (- - - -).

**Figure 9 viruses-15-02335-f009:**
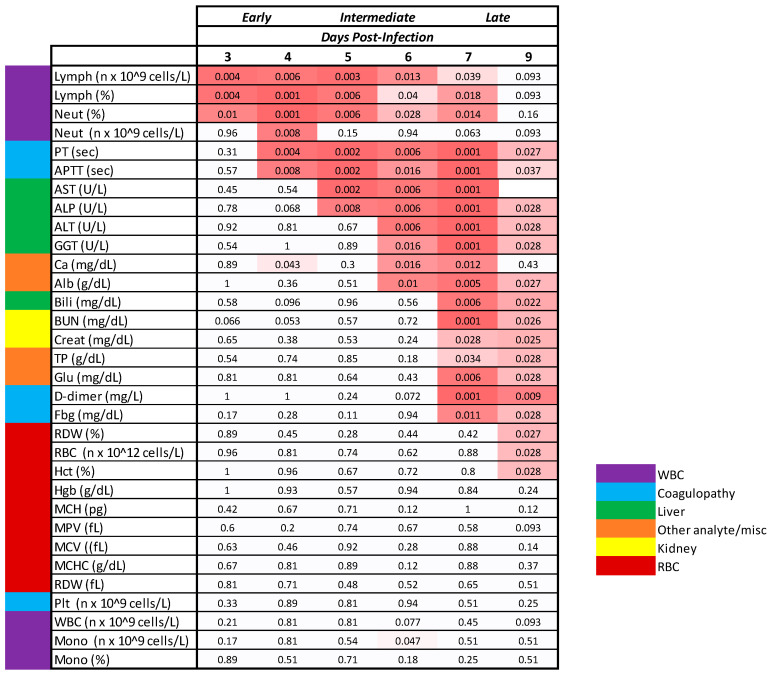
Timing of Statistically Significant Clinicopathologic Changes. The *p*-values shown represent the comparison of MARV-exposed change from baseline to mock-exposed change from baseline. Alb, albumin; Bili, bilirubin; Ca, calcium; Creat, creatinine; Fbg, fibrinogen; Glu, glucose; Lymph, lymphocytes; MCH, mean corpuscular hemoglobin; MCHC, MCH concentration; MCV, mean corpuscular volume; Mono, monocytes; MPV, mean platelet volume; Neut, neutrophils; Plt, platelets; RDW, RBC distribution width; WBC, white blood cells. In the schematic, white cells denote a lack of statistical significance. The pink shading is a heat map with deeper pink denoting greater significance. *p*-Values denoted as 0.001 are either *p* = 0.001 or *p* < 0.001.

**Figure 10 viruses-15-02335-f010:**
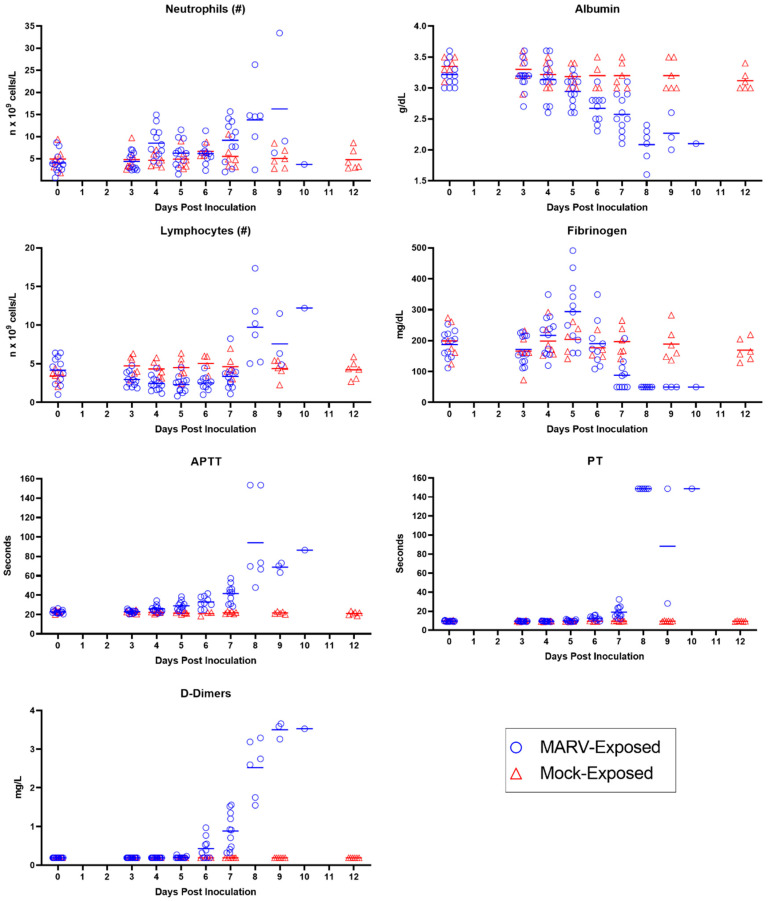
Inflammation and Coagulation Parameters. Measures of inflammation (neutrophils, albumin, and lymphocytes) and coagulation (fibrinogen, APTT, PT, and D-dimers) are depicted for individual MARV-exposed animals (blue circles) and mock-exposed animals (red triangles). Horizontal bars represent means.

**Figure 11 viruses-15-02335-f011:**
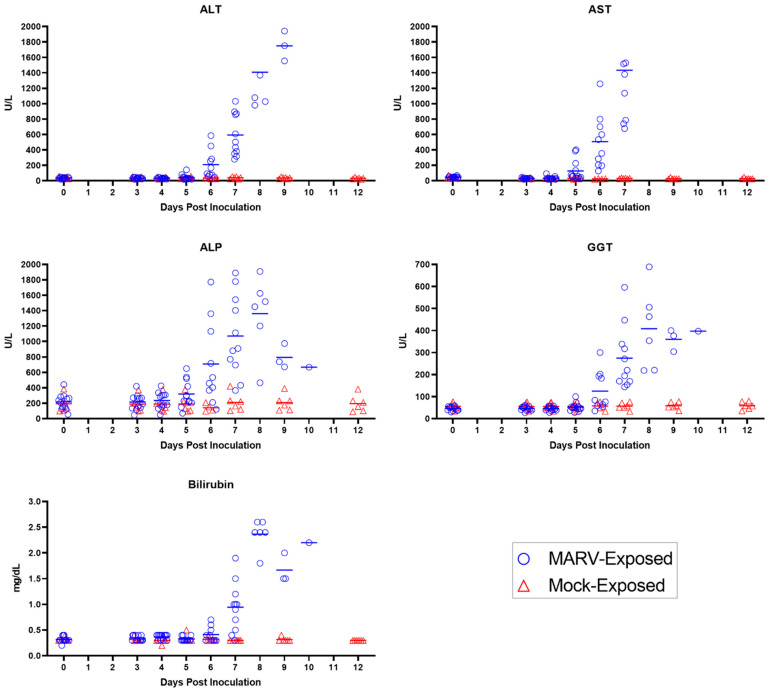
Hepatobiliary Parameters. Measures of hepatobiliary function (ALT, AST, ALP, GGT, and bilirubin) are depicted for individual MARV-exposed animals (blue circles) and mock-exposed animals (red triangles). Horizontal bars represent means.

**Figure 12 viruses-15-02335-f012:**
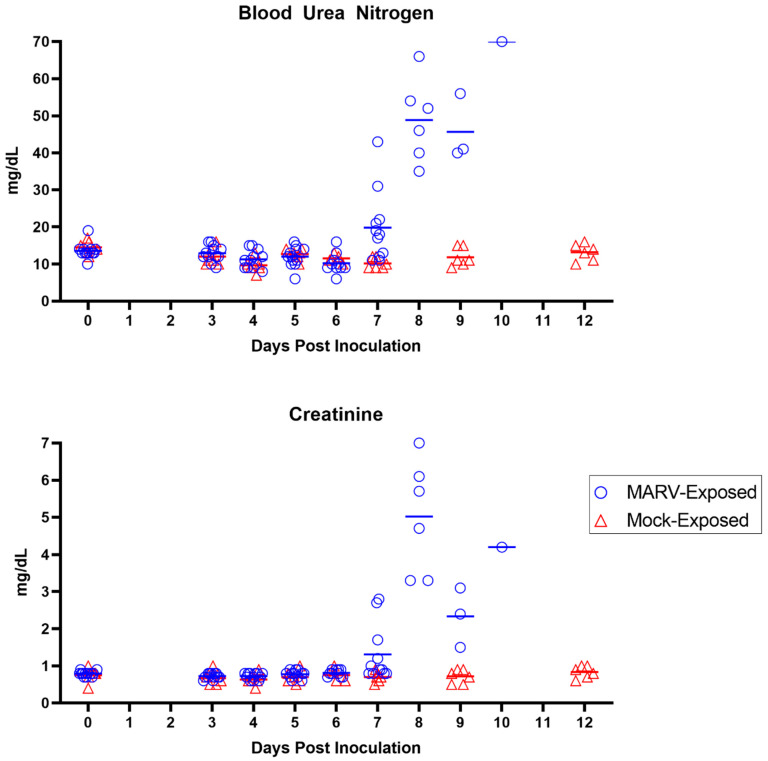
Renal Parameters. Measures of renal function (BUN and creatinine) are depicted for individual MARV-exposed animals (blue circles) and mock-exposed animals (red triangles). Horizontal bars represent means.

**Figure 13 viruses-15-02335-f013:**
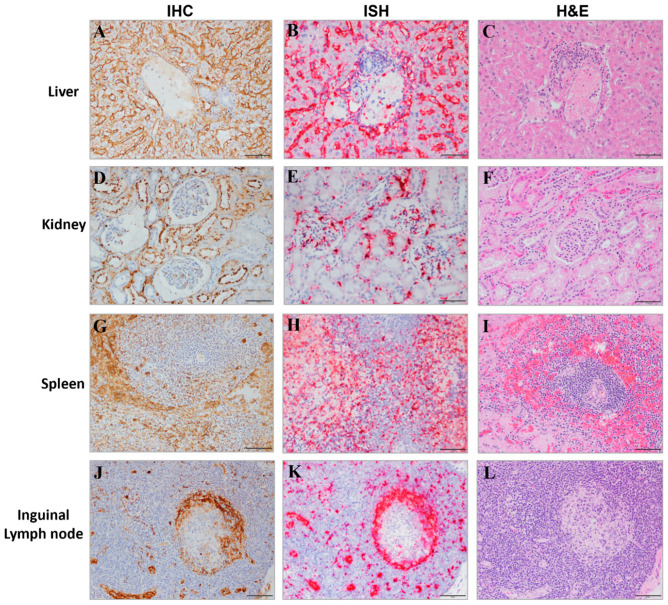
Target Organs of MARV Infection. Representative images of liver (**A**–**C**), kidney (**D**–**F**), spleen (**G**–**I**), and inguinal lymph node (**J**–**L**) tissues from MARV-exposed animals are shown with IHC (left panel), ISH (middle panel) and H&E (right panel) staining from adjacent sections shown. Positive staining from IHC tissue appears as a brown precipitate and the positive ISH staining is red. The scale bar is 50 µm.

**Figure 15 viruses-15-02335-f015:**
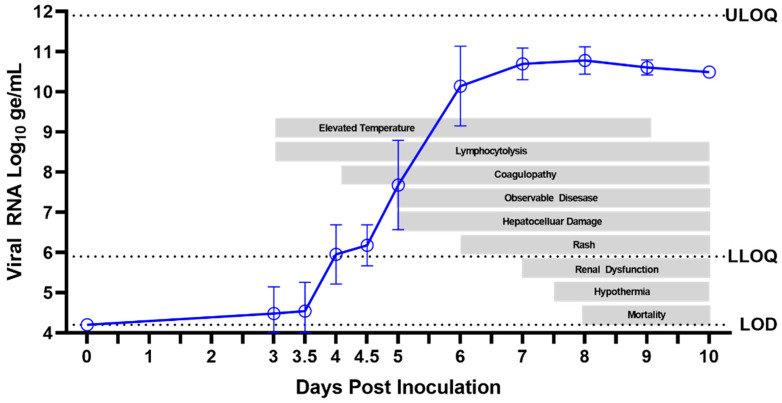
Clinical Progression of Acute MVD in the IM/MARV Cynomolgus Macaque Natural History Study.

**Figure 16 viruses-15-02335-f016:**
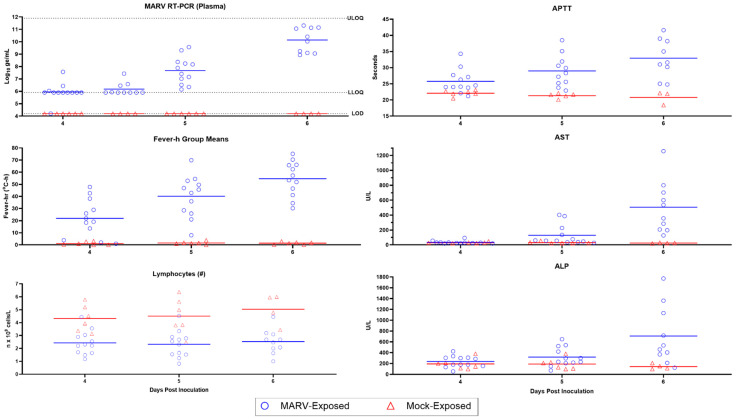
Potential Triggers for Treatment on Days 4–6 PI. Horizontal bars represent means.

**Table 1 viruses-15-02335-t001:** Study Schedule.

Event	Days Pre-Inoculation					Days Post-Inoculation (PI)										
	−9	−8	−7 to −2	−1	0	1–2	3	4	5	6	7	8	9	10	11	12–13 (T ^f^)
Veterinary Release	X															
Move to Containment	X															
Acclimation	X	X	X	X												
Telemetry ^a^	X	X	X	X	X	X	X	X	X	X	X	X	X	X	X	X
MARV Challenge/Mock Exposure					X											
Tether Assembly	X															
Catheter Connection					X											
Awake Observations		X	X	X	X	X	X	X	X	X	X	X	X	X	X	X
Observation (Anesthetized)				X	X											X
Health Status Checks ^b^									X	X	X	X	X			
Plaque Assay					X ^c^		X		X		X		X			X
Polymerase Chain Reaction (PCR)				X	X ^c^		X ^d^	X ^d^	X	X	X		X			X
Hematology					X ^c^		X	X	X	X	X		X			X
Clinical Chemistry					X ^c^		X	X	X	X	X		X			X
Coagulation					X ^c^		X	X	X ^e^	X	X		X			X

^a^ Telemetry baseline data collection started on Day −6 PI. ^b^ Health status checks (i.e., additional cage-side observations) began when animals showed clinical signs (responsiveness score ≥ 1) and were discontinued when all surviving animals had a responsiveness score = 0. ^c^ Day 0 samples were collected prior to challenge. ^d^ Two PCR samples were collected (a.m. and p.m.). ^e^ Two animals (animal 4 from the mock-exposed group, and animal 11 from the MARV-exposed group) weighed less than 3.6 kg on Day 0; therefore, coagulation analysis was not performed on those animals on Day 5 PI. ^f^ T = terminal (moribund, found deceased, or end-of-study). Note: For 7 MARV-exposed animals, terminal blood collection for plaque assay, PCR, hematology, clinical chemistry, and coagulation occurred on Day 8 PI (6 animals) or Day 10 PI (1 animal). Blood was not collected on animals found deceased.

**Table 2 viruses-15-02335-t002:** Survival Time and Demographics.

Group	Animal ID	Sex	Weight at Day 0 (kg)	Age at Day 0 (Years)	Time of Challenge on Day 0	Day PI Deceased	Time Deceased ^c^	Elapsed Time from Challenge to Death	
								(Days)	(Hours)
Mock	1	M	4.1113	5.1	10:38	12	10:43	12.00	288.1
	2	M	3.7624	4.3	10:47	13	09:24	12.94	310.6
	3	F	3.6605	4.3	10:55	12	09:44	11.95	286.8
	4	F	3.4912	4.3	10:59	12	10:25	11.98	287.4
	5	M	4.0521	4.4	11:06	13	09:32	12.93	310.4
	6	M	3.9484	4.6	11:14	13	09:46	12.94	310.5
MARV	7 ^a^	M	4.4746	5.1	11:49	9	01:30	8.57	205.7
	8	M	4.3130	4.2	11:54	9	19:59	9.34	224.1
	9	F	3.6690	3.5	11:59	8	17:11	8.22	197.2
	10	M	4.3382	4.2	12:04	10	07:00	9.79	234.9
	11 ^a^	F	3.4098	4.0	12:11	9	06:00	8.74	209.8
	12	M	3.8370	4.4	12:16	8	08:17	7.83	188.0
	13	M	4.436 ^b^	4.3	12:25	8	08:26	7.83	188.0
	14	F	4.0955	4.3	12:34	8	17:06	8.19	196.5
	15	F	3.9169	4.1	12:42	9	16:17	9.15	219.6
	16 ^a^	F	4.2558	5.1	12:51	8	01:00	7.51	180.1
	17	M	3.7140	4.7	13:06	8	11:42	7.94	190.6
	18	M	4.1870	4.6	13:16	9	00:04	8.45	202.8
**Mean for MARV-Exposed Animals**								8.46	203.1
**Range for MARV-Exposed Animals**								7.51–9.79	180.1–234.9

^a^ Animals 7, 11, and 16 were found deceased. The elapsed time for these three animals is based on the most accurate estimate of the time of death as determined by telemetry (temperature and activity). ^b^ Day 0 weight for animal 13 was recorded only to the thousandth place in raw data. ^c^ Time deceased is the time that death was confirmed after euthanasia except for animals that were found deceased in-cage.

**Table 3 viruses-15-02335-t003:** Responsiveness Score Change in MARV-Infected Animals.

Animal ID	Elapsed Time from Challenge to Responsiveness Score of 1 (Hours)	Elapsed Time from Onset of Responsiveness Score of 1 to Time Deceased ^b^ (Hours)
7 ^a^	131.8	73.9
8	162.2	61.9
9	138.2	59.0
10	114.1	120.9
11 ^a^	162.0	47.9
12	113.9	74.1
13	118.6	69.4
14	137.7	58.9
15	161.5	58.0
16 ^a^	130.9	49.3
17	113.1	77.5
18	137.0	65.8
**Mean**	135.1	68.0
**Range**	113.1–162.2	47.9–120.9

^a^ Animals 7, 11, and 16 were found deceased. The elapsed time for these three animals is based on the most accurate estimate of the time of death as determined by telemetry (temperature and activity). ^b^ Time deceased is the time that death was confirmed after euthanasia (except for animals that were found deceased in cage).

**Table 4 viruses-15-02335-t004:** Potential Triggers for Treatment by Day and Percentage of Animals Affected.

	Day PI										
Parameter (% Impacted)	0	1	2	3	4	5	6	7	8	9	10
Death	0	0	0	0	0	0	0	0	50	92	100
Responsiveness Score ≥ 1	0	0	0	0	0	50	75	100	100	100	100
Body Temp. (Significant Elevation)	0	8	8	50	92	100	100	100	67	33	0
Fever	0	0	0	17	67	92	100	92	50	33	0
RT-PCR (>LOD)	0			25	100	100	100	100	100	100	100
RT-PCR (>LLOQ)	0			0	42	100	100	100	100	100	100
Lymphocyte (↓)				83	92	92	90	73	17	0	0
APTT (↑)				—	92	100	100	100	100	100	100
AST (↑)				—	—	75	100	100	100		
ALP (↑)				—	—	92	100	100	100	100	100

Orange shading denotes the percentage of animals impacted out of those tested for each parameter each day. A dash (—) indicates that the percentage of animals was not assessed because the *p*-value for change from baseline comparing MARV- to mock-exposed animals was not statistically significant at these times.

**Table 5 viruses-15-02335-t005:** Comparison of Three MARV Natural History Studies in Cynomolgus Macaques.

Parameter	Zumbrun et al.	Alfson et al. [28]	Comer et al. [30]
Quality System	GLP	A quality system consistent with GLP	Well-documented/controlled
Cynomolgus Origin	Cambodian	Chinese	Asiatic, bred in Vietnam
Animal Age (years)	3.5–5.1	3.39–4.10	2.5–3.2
Animal Weight (kg)	3.41–4.47	2.55–3.77	2.4–3.2
Challenge Stock	P3 (USAMRIID in-house stock)	P3 (UTMB in-house stock)	P2 (BEI Resources)
Challenge Dose (Target; Actual)	1000 pfu; 1125 pfu	1000 pfu; 222–360 pfu	1000 pfu; 6500–8000 pfu
n = MARV, n = Mock	12 MARV; 6 mock	8 MARV; 2 mock ^a^	12 MARV; 6 mock
#Males vs. #Females	11 males, 7 females	5 males; 5 females	9 males; 9 females
Survival	8.56 days (mean)	7.26 (mean ^b,c^)	8.13 days (mean ^b^)
	8.33 days (median)	7.95 days (Kaplan–Meier median)	7.31 days (median)
	7.51–9.79 days (range)	7–9 days (range ^c^)	6.43–7.98 days (range)
	d8 (n = 5), d9 (n = 6), d10 (n = 1)	d7 (n = 1), d8 (n = 5), d9 (n = 2)	d6 (n = 1), d7 (n = 8), d8 (n = 3)
Euthanasia Criteria	Scoring system with a single parameter (responsiveness), with euthanasia triggered by a score of 4.	Scoring system with 13 parameters, including: food/enrichment/fluid consumption, stool, dehydration, appearance (rough hair/coat), nasal discharge, bleeding, body weight, rectal temperature, petechia, labored/agonal breathing, and responsiveness. Euthanasia triggered by total score ≥ 15, or combination between responsiveness score and temperature change and/or chemistry changes above a predetermined threshold.	Score system with 5 parameters, including: respiration, food consumption, feces/urine, activity/appearance, bleeding/hemorrhage/rash. Euthanasia triggered by a score of ≥10.
Observation Frequency	up to 5× daily	up to 4× daily	2–3× daily
Blinding	No	Yes (veterinary technicians, veterinarians, in vitro staff, necropsy staff, pathologist)	No
Catheters vs. Sedated for Blood Collection	Catheters ^d^	Sedation	Sedation
Blood Collection Timepoints (Days PI)	0, 3 (a.m.), 3 (p.m.), 4 (a.m.), 4 (p.m.), 5, 6, 7, 8, 9, 10, 12	−7, 0, 2, 3, 5, 7, 9, 11, 13, 14	−4, 0, 3, 5, 7, 10, 14, 21, 28
Telemetry	DSI-M00; temperature and activity	DSI-M00; temperature and activity	DST Micro-T Star Odi; temperature
RT-PCR	plasma	serum	serum
Plaque Assay	serum	serum	serum
CBC	yes	yes	yes
Serum Chemistry	yes	yes	yes
Coagulation	yes	yes	yes
Immunological Profile	no	yes (cytokine and chemokine)	no
IgG ELISA	no	no	yes (no + samples)
Anatomical Pathology	yes	yes	yes
Histology	yes	yes	yes
IHC	yes	no	no
ISH	yes	no	no
Tissue Viral Load	no	yes	no

^a^ Alfson et al. included an arm of the study for scheduled euthanasia. Only animals from the unscheduled euthanasia arm of the study are included in this comparison. ^b^ Mean was not reported in Alfson et al. or Comer et al. This mean is based on our calculation using the published survival datapoints. ^c^ In Alfson et al., only whole number days for survival were published and the mean and range in this table is based on those reported survival datapoints. ^d^ Six animals on one or more occasions each were anesthetized for blood collection when catheters were nonpatent or for replacement of damaged jackets (two animals on one occasion each). Animals were not anesthetized on more than three consecutive days or when assigned a responsiveness score ≥ two.

## Data Availability

The data presented in this study are available on request from the corresponding author and with permission from Gilead Sciences and JPEO-CBRND.

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
