# Peer review of "Characterization of the Cynomolgus Macaque Model of Marburg Virus Disease and Assessment of Timing for Therapeutic Treatment Testing"

_viruses, 2023, doi:10.3390/v15122335_

Round 1

Reviewer 1 Report

Comments and Suggestions for Authors

In their manuscript entitled “Characterization of the Cynomolgus Macaque Model of Marburg Virus Disease and Assessment of Timing for Therapeutic Treatment Testing,” Zumbrun et al. provide an exceptionally thorough description of Marburg virus disease in a common filovirus animal model, Cynomolgus macaques. The authors should be commended for conducting a challenging experiment under Good Laboratory Practices and for providing extensive experimental details, results, and conclusions. This is a valuable piece of work that not only improves our understanding of Marburg virus disease but also enhances the value of the Cynomolgus macaque model. The analysis of triggers to treat are particularly useful. In general, this article is very well written and organized. I have only very minor comments/suggestions, listed below:

-       - Have the authors deposited the genome sequence of their Marburg virus isolate into Genbank? Can they provide an accession number?

-       - The authors state that a validated RT-qPCR method was used to quantify viral RNA in plasma. Can the authors please provide a citation that describes this method? Alternatively, can the authors provide more details about their RT-qPCR assay, including the composition of the master mix, primer sequences, and thermal cycler run parameters? Can the authors also describe the formula they used for calculating genome equivalents?

-      -  Very minor, but it would be ideal if the x-axes in Figure 3 were identical.

Reviewer 2 Report

Comments and Suggestions for Authors

The Marburg virus (MARV) causes severe, often fatal, human disease, with outbreaks linked to natural hosts such as the Egyptian fruit bat. There are no licensed MARV vaccines or treatments, so animal models are required for countermeasure development. The manuscript titled “Characterization of the Cynomolgus Macaque Model of Marburg Virus Disease and Assessment of Timing for Therapeutic Treatment Testing” aimed to investigate the disease manifestations and pathogenesis of Marburg virus (MARV) infection in cynomolgus macaques. In this study, a total of twelve macaques were exposed to MARV, all of them became infected, exhibited signs of illness on Days 8-10 post-inoculation. Fever and other clinical symptoms began to appear on Day 4 post-inoculation, with reduced responsiveness in some animals observed on Day 5.

The study found that systemic inflammation, coagulopathy, and direct cytopathic effects of MARV contributed to multiorgan dysfunction and, ultimately, the demise of all infected macaques.

The authors conclude that fever, systemic viremia, lymphocytolysis, coagulopathy, and hepatocellular damage could serve as indicators for initiating treatment in future studies evaluating potential therapeutic interventions for MARV infection.

Overall, the study is carefully designed and carried out, the manuscript is well-written. The results advance our understanding of Marburg virus pathogenesis, provide valuable insights into disease progression in cynomolgus macaques, and provide a foundation for developing and testing potential therapeutic interventions.

Author Response

We would like to thank the reviewers for the positive and encouraging feedback on our manuscript entitled “Characterization of the Cynomolgus Macaque Model of Marburg Virus Disease and Assessment of Timing for Therapeutic Treatment Testing.” We appreciate the comments and hope that the manuscript will now be ready for publication in Viruses.

Reviewer 3 Report

Comments and Suggestions for Authors

This study by Zumbrun et al details Marburg disease in cynomolgus macaques. The well written study follows the animals through the course of infection, carefully monitoring disease signs, numerous biologic parameters, and viral genome and viremia levels. The data is clearly presented and highlights specific parameters to focus on in future studies. The work is also compared to similar studies. I am confident the study will be well cited as a baseline for Marburg disease and is an important study for future work on potential therapeutics and vaccines.

Author Response

(The authors gave the same response as above.)
